# Mouse fetal growth restriction through parental and fetal immune gene variation and intercellular communications cascade

Gurman Kaur [1,2,17], Caroline B. M. Porter[2,17], Orr Ashenberg[2], Jack Lee[3], Samantha J. Riesenfeld [2,13,14], Matan Hofree [2], Maria Aggelakopoulou[4], Ayshwarya Subramanian [2], Subita Balaram Kuttikkatte[1], Kathrine E. Attfield [4], Christiane A. E. Desel [4,15], Jessica L. Davies [4], Hayley G. Evans[4], Inbal Avraham-Davidi [2], Lan T. Nguyen[2], Danielle A. Dionne[2], Anna E. Neumann [5], Lise Torp Jensen[6], Thomas R. Barber[1], Elizabeth Soilleux [7], Mary Carrington [8,9], Gil McVean [10], Orit Rozenblatt-Rosen [2,16], Aviv Regev[2,11,12,16] ✉ & Lars Fugger [1,4,6] ✉

Fetal growth restriction (FGR) affects 5–10% of pregnancies, and can have serious consequences for both mother and child. Prevention and treatment are limited because FGR pathogenesis is poorly understood. Genetic studies implicate *KIR* and *HLA* genes in FGR, however, linkage disequilibrium, genetic influence from both parents, and challenges with investigating human pregnancies make the risk alleles and their functional effects difficult to map. Here, we demonstrate that the interaction between the maternal KIR2DL1, expressed on uterine natural killer (NK) cells, and the paternally inherited HLA-C*0501, expressed on fetal trophoblast cells, leads to FGR in a humanized mouse model. We show that the KIR2DL1 and C*0501 interaction leads to pathogenic uterine arterial remodeling and modulation of uterine NK cell function. This initial effect cascades to altered transcriptional expression and intercellular communication at the maternal-fetal interface. These findings provide mechanistic insight into specific FGR risk alleles, and provide avenues of prevention and treatment.

[1]MRC Human Immunology Unit, MRC Weatherall Institute of Molecular Medicine, John Radcliffe Hospital, University of Oxford, Oxford, UK. [2]Klarman Cell Observatory, Broad Institute of MIT and Harvard, Cambridge, MA, USA. [3]Department of Biomedical Engineering, School of Biomedical Engineering and Imaging Sciences, King's College London, London, UK. [4]Oxford Centre for Neuroinflammation, Nuffield Department of Clinical Neurosciences, MRC Weatherall Institute of Molecular Medicine, John Radcliffe Hospital, University of Oxford, Oxford, UK. [5]Broad Institute of MIT and Harvard, Cambridge, MA, USA. [6]Department of Clinical Medicine, Aarhus University Hospital, Aarhus, Denmark. [7]Department of Pathology, Tennis Court Rd, University of Cambridge, Cambridge, England. [8]Basic Science Program, Frederick National Laboratory for Cancer Research in the Laboratory of Integrative Cancer Immunology, National Cancer Institute, Bethesda, MD, USA. [9]Ragon Institute of MGH, MIT, and Harvard, Cambridge, MA, USA. [10]Big Data Institute, Li Ka Shing Centre for Health Information and Discovery, University of Oxford, Oxford, UK. [11]Massachusetts Institute of Technology, Department of Biology, Cambridge, MA, USA. [12]Howard Hughes Medical Institute, Chevy Chase, MD, USA. [13]Present address: Pritzker School of Molecular Engineering, University of Chicago, Chicago, IL, USA. [14]Present address: Department of Medicine, University of Chicago, Chicago, IL, USA. [15]Present address: University Department of Neurology, University Hospital Magdeburg, Magdeburg, Germany. [16]Present address: Genentech, 1 DNA Way, South San Francisco, CA, USA. [17]These authors contributed equally: Gurman Kaur, Caroline B. M. Porter. ✉e-mail: aregev@broadinstitute.org; lars.fugger@ndcn.ox.ac.uk

Fetal growth restriction (FGR) refers to pathologically reduced fetal growth, which occurs in about 5–10% of pregnancies worldwide[1–3]. FGR is one of the largest contributors to perinatal mortality and morbidity[4–9], and predisposes individuals to heart disease, hypertension, type 2 diabetes, and stroke later in life[10–16]. It can occur alone or with pre-eclampsia, a serious and systemic pregnancy complication characterized by maternal new-onset hypertension and proteinuria[17,18].

Despite their serious clinical implications, the causes, disease mechanisms and relationship of FGR and pre-eclampsia are poorly understood, and effective prevention and treatment strategies are lacking[19–21]. This is partly because studying FGR and pre-eclampsia in human pregnancies is logistically challenging, due to the inaccessibility of tissue during critical early stages of pregnancy. Thus, there is an unmet need to better understand disease pathogenesis to identify diagnostic methods and therapeutic interventions that improve maternal and fetal health, and long-term health of the child.

FGR risk factors can be fetal, maternal or placental in origin. Pathogenesis is thought to occur early in the formation of the placenta, known as placentation, and is evidenced by placental insufficiency and dysfunction, such as higher uterine resistance to blood flow[17,19,22,23]. Genetic association studies in humans suggest that uterine natural killer (uNK) cell surface receptors and trophoblast ligands are involved in controlling placentation. uNK cells express a combination of inhibitory and activating killer cell immunoglobulin-like receptors (KIRs) on their cell surface in humans, are the most abundant lymphocytes at the maternal–fetal interface, especially during early to mid-gestation, and are implicated in placental vascular remodeling[24–30]. Human leukocyte antigen-C (*HLA-C*) alleles encode trophoblast KIR ligands and can modulate NK cell function. HLA-C is the only polymorphic HLA molecule expressed on the surface of fetal extravillous trophoblasts, which are cells derived from the outer layer of the blastocyst and define the boundary between the mother and the fetus. There is direct contact between uNK and trophoblast cells during placentation[31–34].

Genetic studies suggest that pregnant women with two copies of the *KIR A* haplotype (i.e., *KIR AA*), which is composed of inhibitory *KIR* genes, have a higher risk of pregnancy complications such as FGR, pre-eclampsia and recurrent miscarriage, specifically, if the fetus inherits a group 2 *HLA-C* allele from the father (odds ratio 2.02), or when the fetus has more group 2 *HLA-C* alleles than the mother (odds ratio 2.09)[30,35–37], however, association with a specific *HLA-C* allele has not been defined. In contrast, maternal presence of the *KIR B* haplotype, which is mostly composed of activating *KIR* genes, is associated with protection against pregnancy disorders and high birth weight[30,35–37]. Polymorphic *HLA-C* alleles are classified into two groups based on amino acid dimorphism at position 80 (Group 1—$^{80}$Asn or Group 2—$^{80}$Lys), which defines their ability to bind specific KIR2D receptors. Group 1 *HLA-C* alleles include *HLA-C*01, C*03, C*07, C*08*, among others, and Group 2 *HLA-C* alleles include *HLA-C*02, C*04, C*05, C*06, C*0707/9, C*1204/5, C*15, C*1602, C*17* and *C*18*[38–40]. Both *KIR* and *HLA* genes are highly polymorphic and inherited in complex haplotypes[26,33,41], making it genetically challenging to ascertain the function of specific genes in disease pathogenesis, therefore, to determine the function of specific genes, one needs to tease them apart from their haplotype and identify their effects by functional analyses.

Here, we develop an humanized mouse model system and show that products of two risk alleles, specifically, the inhibitory KIR2DL1 receptor—a *KIR A* haplotype gene—on maternal uNK cells, and paternally derived HLA-C*0501—an HLA-C group 2 allotype—on fetal trophoblast cells, leads to FGR. Our system clarifies this genetic association and highlights the functional effects of this specific KIR-HLA-C interaction in disease pathogenesis, allowing us to study its function at the maternal–fetal interface in pregnancy and provide insights into the mechanisms underlying KIR2DL1:HLA-C-induced FGR.

## Results

### A humanized HLA-C*05 and KIR2DL1 transgenic mouse model

In order to disentangle the genetic association of the *KIR A* haplotype and group 2 *HLA-C* alleles with FGR, we focused on two common *KIR A* haplotype and *HLA-C* group 2 alleles because FGR is a common pregnancy complication with high disease incidence. We reasoned that *KIR2DL1* is a candidate *KIR A* haplotype gene likely to confer genetic risk to FGR because it is always present on the risk *KIR A* haplotype and has strict binding specificity to group 2 *HLA-C* alleles; *KIR2DL1*0030201* is one of the most common *KIR2DL1* inhibitory alleles typically found on the *KIR A* haplotype[26]. Correspondingly, we chose *HLA-C*05* (*HLA-C*05:01:01:01*), which is a common group 2 *HLA-C* allele in the Caucasian population[42], and has been shown to bind KIR2DL1[38]. Although the associations seen at a population level among the respective allelic groups may not apply to any specific pair of alleles, by choosing common alleles from both gene groups, we increase the chances of this being disease relevant. Therefore, we developed humanized single transgenic mice with physiological expression of HLA-C*05 or KIR2DL1 and double transgenic mice expressing both HLA-C*05 and KIR2DL1.

The HLA-C*05 transgenic mice expressed HLA-C*05 on all MHC class I expressing cells as HLA-C*05 was driven by its endogenous *HLA-C*05* promoter. To facilitate interaction of the HLA-class I molecule with murine CD8 (and hence mouse T-cell receptors), we replaced the α3 domain of HLA-C*05 with its murine counterpart from H2-K$^b$, along with the adjacent transmembrane and cytoplasmic domains—as has been done for other HLA-Class I genes[43] (Fig. 1a). We confirmed that HLA-C*05 was expressed in all cells of all tested organs (Fig. 1b and Supplementary Fig. 1a), and in different populations of immune cells in the spleen or thymus, and thymic epithelial cells (TECs) (Fig. 1c, d), demonstrating physiological expression of the transgene in the mouse model. HLA-C*05-expressing splenocytes responded to in vitro stimuli such as IFNγ or LPS by upregulating HLA-C expression (Fig. 1e, f and Supplementary Fig. 1b, c), in agreement with previous observations[44].

To model the fact that *KIR* genes are predominantly expressed on NK cells in humans[26,45], we generated mice where KIR2DL1 is restricted to murine NK cells. To this end, we generated transgenic mice with a loxP flanked stop cassette between the *KIR2DL1* coding DNA sequence and CAG promoter, and mated them to mice expressing iCre-recombinase in cells expressing the NK cell marker, *Ncr1* (i.e., *Ncr1-iCre* BAC transgenic mice)[46] (Fig. 1g), to generate KIR2DL1-expressing mice. Staining immune cells using anti-KIR antibody demonstrated specific and high expression of KIR2DL1 on NK cells only, with expression in ~75–90% of NK cells (Fig. 1h, i).

Expression of KIR2DL1 or HLA-C*05 did not alter the proportion of NK cells (Supplementary Fig. 1d) or other immune cell types (Supplementary Fig. 1e, f), NK cell receptor and transcription factor expression (Supplementary Fig. 1g–i), and development and maturation of NK cells tested by CD27 and CD11b expression[47], remained unaltered in the transgenic mice (Fig. 1j, k and Supplementary Fig. 1j, k). Engineering of *KIR* and *HLA* genes in this transgenic model was a superposition of function, i.e., we did not knockout any mouse MHC class I or NK cell receptor molecules or abrogate any immune cell types. Expectedly, we did not see overall systemic effects as a result of the introduced transgenes.

In summary, to test the hypothesis that KIR2DL1 and HLA-C*05 in combination lead to the development of FGR, we developed an humanized transgenic model system with specific expression of the inhibitory KIR2DL1 receptor on murine NK cells and the corresponding expression of the KIR2DL1 ligand, HLA-C*05, on all MHC class I expressing cells in mice.

### KIR2DL1 and HLA-C*05 are functional in the transgenic mice

To test whether signaling via the inhibitory KIR2DL1 receptor impacted NK cell responses, we stimulated splenocytes from WT,

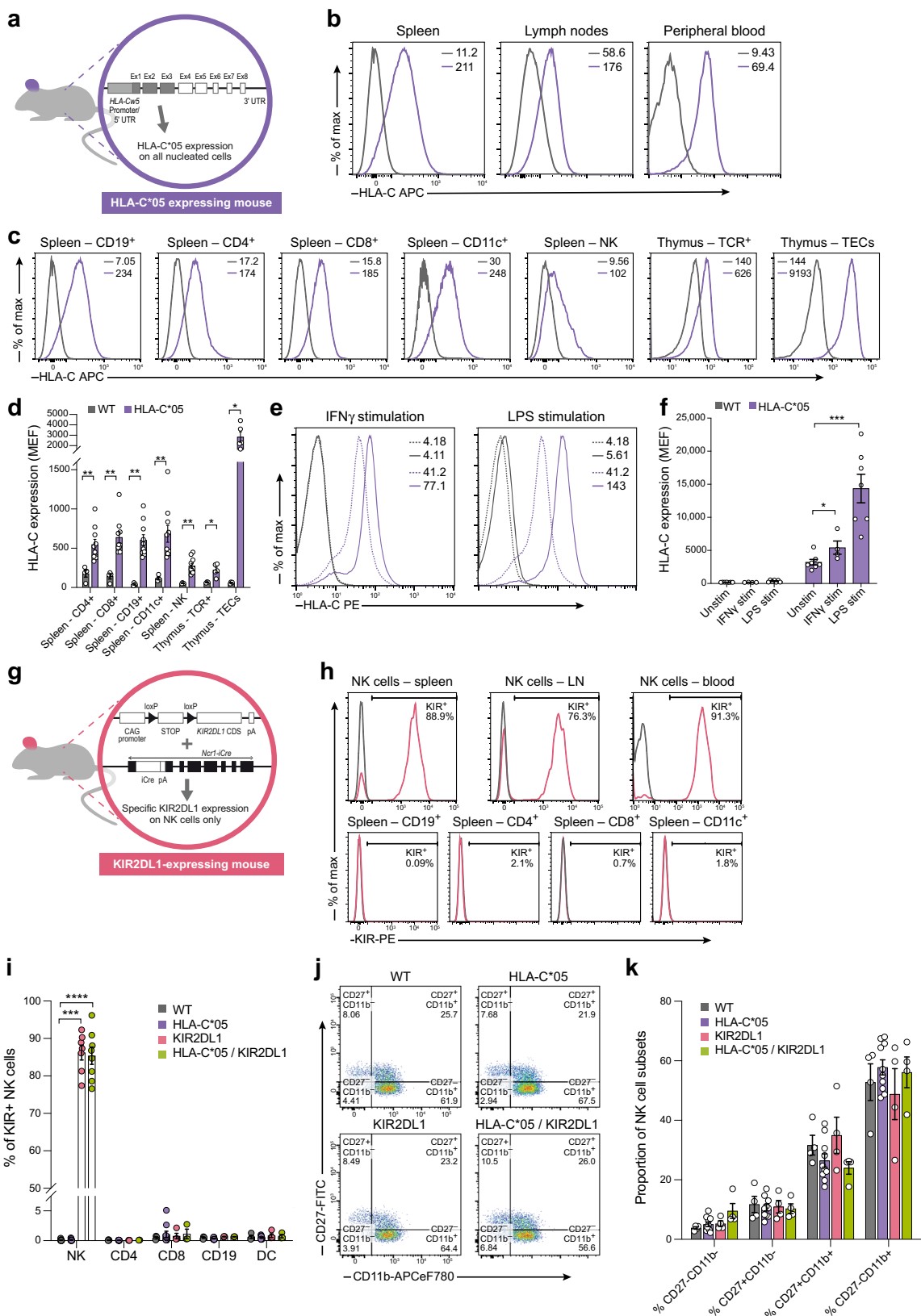

HLA-C*05 or KIR2DL1-expressing transgenic mice in vitro using antibodies that activate NK cell receptors (NKp46 or Ly49D), observing the expected increase in IFNγ production by NK cells. Presence of α-KIR2DL1 antibody along with α-NKp46 or α-Ly49D led to significant decrease (p-value < 0.05, Mann–Whitney U-test) in IFNγ production in KIR2DL1-expressing splenocytes, but not in HLA-C*05 or WT splenocytes (Fig. 2a, b), confirming inhibitory signaling via the KIR2DL1 transgene.

We also assessed KIR2DL1 binding by HLA-C*05 in vitro by incubating the KIR2DL1-expressing NK cell line (YT-KIR2DL1) with 721.221-C*05 transfectants. Co-culture with 721.221-C*05 transfectants, but not the control 721.221-vector or 721.221-C*07 transfectants, dampened

**Fig. 1 | HLA-C*05 and KIR2DL1 transgenic mouse model. a** Schematic of construct used to make HLA-C*05 transgenic mice. Gray shaded region: *HLA-C* allele, white shaded region: murine *H-2K^b* allele. **b, c** Representative cell surface expression of HLA-C*05 on total cells from organs (**b**) and on gated cell subsets from spleen (**c**). NK cells are NKp46⁺CD3⁻TCR⁻, thymus TCR + cells are CD3⁺TCR⁺ and TECs are Epcam⁺CD45⁻ cells. Numbers denote mean florescence intensity (MFI). **d** HLA-C cell surface expression plotted as molecules of equivalent fluorochrome (MEF). **e** Representative cell surface HLA-C*05 expression on spleen CD19⁺ cells cultured with (solid lines) or without stimulation (dashed lines). Numbers denote MFI. **f** HLA-C cell surface expression plotted as MEF from CD19⁺ splenocytes. **g** Schematic of *KIR2DL1* and *Ncr1-iCre* gene expression construct used to make NK cell-specific KIR2DL1-expressing mice. **h** Representative cell surface expression of KIR2DL1 on NKp46⁺CD3⁻TCR⁻ NK cells in different organs or on different cell subsets from spleen. Numbers denote percentage of KIR⁺ cells. **i** Percentage of KIR⁺ cells in immune cell subsets from the spleen. **j, k** Cell surface

staining of CD27 and CD11b on gated splenic NK cells. Purple: HLA-C*05 mice, gray: WT mice, pink: KIR2DL1-expressing mice, green: HLA-C*05/KIR2DL1 double transgenic mice. Mean ± SEM is shown. *n* represents biologically independent animals in each group. **d** *n* = 4 (WT), 10 (HLA-C*05), for Spleen−CD4⁺, CD8⁺, CD19⁺, CD11c⁺, and NK cells. *n* = 3 (WT), 5 (HLA-C*05), for Thymus-TCR⁺. *n* = 4 (WT), 5 (HLA-C*05), for Thymus-TECs. **f** *n* = 7 (Unstim), 4 (IFNγ stim) and 7 (LPS stim). **i** *n* = 9 (WT), 10 (HLA-C*05), 7 (KIR2DL1), 8 (HLA-C*05, KIR2DL1) for NK; *n* = 4 (WT), 10 (HLA-C*05), 4 (KIR2DL1), 3 (HLA-C*05,KIR2DL1) for CD4, CD8, CD19, and DC. **k** *n* = 4 (WT), 10 (HLA-C*05), 4 (KIR2DL1), and 4 (HLA-C*05,KIR2DL1) for all conditions. **p* < 0.05, ***p* < 0.01, ****p* < 0.001, *****p* < 0.0001, Two-tailed Mann−Whitney *U*-test. Exact *p*-values are: **d** HLA-C*05 vs. WT−*p* = 0.0357 (Thymus TCR + ), *p* = 0.0159 (Thymus TECs), *p* = 0.002 (all others). **f** *p* = 0.0424 (IFNγ stim vs. Unstim), *p* = 0.0006 (LPS stim vs. Unstim). **i** *p* = 0.0002 (KIR2DL1 vs. WT), *p* < 0.0001 (HLA-C*05/KIR2DL1 vs. WT). Source Data are provided as a Source Data file.

the production of IFNγ by KIR2DL1-expressing NK cell line, confirming that HLA-C*05 (but not HLA-C*07 or the empty-vector transduced cells) was bound by KIR2DL1 (Supplementary Fig. 1l).

Next, to assess whether the KIR2DL1 transgene conferred inhibitory signals to NK cells upon recognition of HLA-C*05 in vivo, we performed adoptive transfer experiments using splenocytes from either knockout mice of murine MHC class I molecules, H2K^b and H2D^b ("KO cells"), WT mice with normal expression of H2K^b and H2D^b ("WT cells"), or HLA-C*05 transgenic mice bred to H2K^b and H2D^b KO ("HLA-C*05⁺ KO cells"). We labeled KO, WT and HLA-C*05⁺ KO splenocytes with different concentrations of the florescent cell labeling dye CFSE, and injected them into either WT or KIR2DL1-expressing transgenic mice. As expected, KO cells showed increased rejection in both WT or KIR2DL1-expressing recipient mice due to missing-self recognition, given the capacity of NK cells to attack cells that do not express sufficient levels of MHC class I molecules of the host[48]. The HLA-C*05⁺ KO cells were protected from NK cell-mediated rejection in KIR2DL1-expressing mice (but not WT mice) (*p*-value < 0.001, Mann−Whitney *U*-test) (Fig. 2c−e), showing that HLA-C*05 engagement by the KIR2DL1-expressing NK cells repressed mouse NK cell activating pathways, confirming the functionality of the humanized model.

## KIR2DL1 and HLA-C*05 are expressed at the maternal−fetal interface

We next evaluated expression of the *KIR2DL1* and *HLA-C*05* transgenes at the maternal−fetal interface in two different mating combinations: (1) KIR2DL1-expressing female mated with a HLA-C*05-expressing male (KIR × C*05), chosen because in humans there is a higher risk of FGR when the mother has the *KIR A* haplotype (carrying KIR2DL1) and the father contributes the group-2 *HLA-C* allele[30,35–37]; and (2) WT female mated with a HLA-C*05-expressing male (WT × C*05), as a control. HLA-C*05 mating males were homozygous for HLA-C*05, such that all fetuses in the litter would have the paternally derived *HLA-C*05* allele. As the frequency of uNK cells peaks at early to mid-gestation and declines in late pregnancy[24,25], we isolated uNK cells from mid-gestation (gestation day (gd) 9.5) from the maternal−fetal interface of KIR × C*05 and WT × C*05 matings.

In KIR × C*05, KIR2DL1 was expressed in ~85−90% of all uNK cells−including CD49⁺ or DX5⁺ cell subsets, known to distinguish tissue resident (trNK) and conventional (cNK) NK cells (cNK cells circulate in the blood and spleen), respectively[49]; no KIR staining was observed in WT × C*05 (Fig. 2f, g). A third control, where a double transgenic female expressing HLA-C*05 and KIR2DL1 was mated with a WT male (C*05/KIR × WT), also expressed KIR in ~85−90% of all uNK cells (Fig. 2g). The proportion of total uNK cells or CD49a⁺ and DX5⁺ uNK cell subsets was comparable across mating combinations (Supplementary Fig. 1m, n).

We assessed expression of HLA-C*05 at the maternal−fetal interface in uterine tissue isolated from implantation sites at gd9.5 or

gd10.5 of an HLA-C*05 negative female mated to either an HLA-C*05 positive or HLA-C*05 negative male. This meant that HLA-C*05 expression at the maternal−fetal interface could be ascribed only to the paternally derived *HLA-C*05* allele on fetal trophoblast cells. We detected *HLA-C* mRNA expression in C*05⁺ but not in C*05⁻ fetal cells (Fig. 2h), and *HLA-C* DNA, by genotyping of the fetus using genomic DNA isolated from embryonic tissue. Furthermore, we isolated and cultured fetal trophoblast cells from gd12.5 placentas and validated the paternally derived HLA-C expression in HLA-C*05⁺ trophoblast cells by flow cytometry. As previously shown for trophoblast cells[31], in vitro stimulation with LPS + IFNγ led to a significant upregulation of HLA-C expression in HLA-C*05⁺ (*p*-value < 0.01, Mann−Whitney *U*-test), but not HLA-C*05⁻, trophoblast cells (Fig. 2i, j).

These results demonstrated the expression of the *KIR2DL1* and *HLA-C*05* transgenes at the maternal−fetal interface, confirming the validity of this model to study FGR-relevant disease mechanisms.

## Maternal KIR2DL1 and paternal HLA-C*05 expression leads to FGR

As birth and fetal weights are markers for intrauterine development, we investigated the effect of the KIR-HLA-C interactions in mediating FGR by assessing fetal weight in the three mating combinations described above (KIR × C*05, WT × C*05, and C*05/KIR × WT), as well as two additional controls (KIR × WT and WT × WT). Fetal weight was measured just before birth, at gd18.5, to avoid milk uptake as a confounding factor.

Pregnancies involving females with KIR2DL1 expression on uNK cells and males with expression of HLA-C*05 (KIR × C*05 mating) led to FGR in the fetuses, evidenced by a significant reduction in the average fetal weight of the embryos compared to all other mating combinations. The progeny from the KIR × C*05 mating were -12% lower in weight compared to the control WT × C*05 mating (Fig. 3a, *p*-value < 10⁻³, linear mixed-effects model) and -56% of KIR × C*05 fetuses were below the 10th weight percentile of control WT × C*05 fetuses (with 36% below the 5th percentile of control WT × C*05 fetuses) (Fig. 3b). The proportion of KIR ×C*05 fetuses that were below the 5th or the 10th weight centile of the control were statistically indistinguishable between male and female fetuses (*p* > 0.05, Mann−Whitney *U*-test). The placental weight (also measured at gd18.5) was very similar across mating combinations (Fig. 3c), concordant with the lack of observed lesions or pathology in the placenta assessed by H&E staining. We did not observe any difference in litter sizes between the different mating groups (*p* > 0.05, ordinary one-way ANOVA). The observed 12% reduction in average fetal weight and the skewed fetal weight distribution of the KIR × C*05 fetuses is in line with clinical observations observed in FGR[8,21,37].

Pregnant KIR2DL1-expressing females from the KIR × C*05 mating did not show evidence of hypertension throughout the course of gestation and post-partum (Supplementary Fig. 2a) or of proteinuria (Supplementary Fig. 2b), both hallmarks of pre-eclampsia[18]. We also

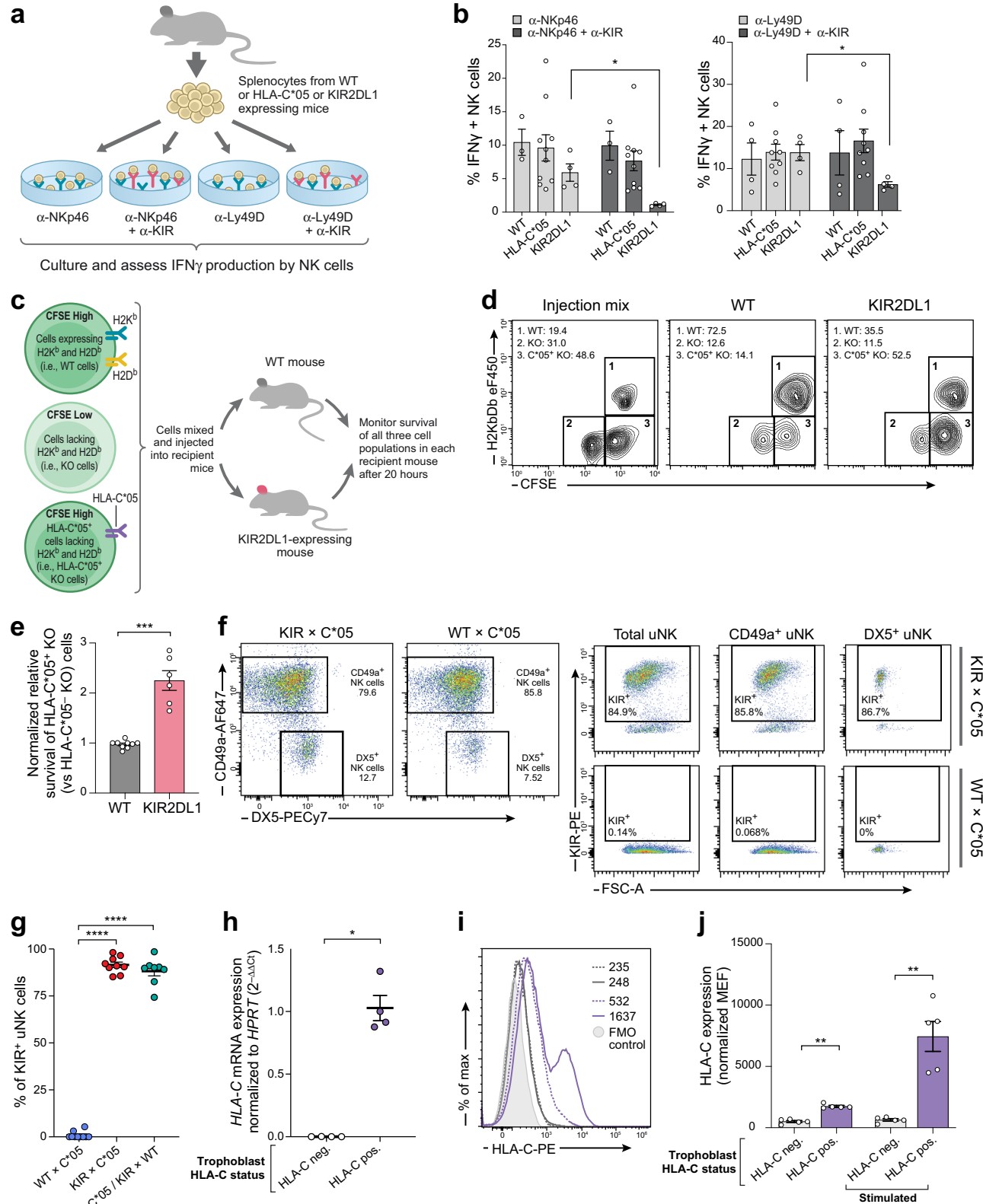

did not observe any change in circulating levels of molecules associated with an increased risk of developing pre-eclampsia[50–52], such as soluble fms-like tyrosine kinase 1 (sFLT-1), soluble Endoglin or placental growth factor (PlGF) in the plasma of pregnant females from the KIR × C*05 mating (Supplementary Fig. 2c–e).

Collectively, these results demonstrate that the parental combination of *KIR2DL1* and *HLA-C*05* genes lead to FGR. The

impact we observed on growth restriction in fetuses was not associated with any manifestations of a systemic disease such as pre-eclampsia, perhaps because pre-eclampsia is a disease of the human species and does not seem to occur in most other mammals[53]; however, it is also possible that manifestation of pre-eclampsia requires additional risk factors or mechanisms on top of those involved in FGR.

**Fig. 2 | *KIR2DL1* transgene recognizes HLA-C*05, modifies NK cell function.**
**a** Experimental design of NK cell stimulation experiments. **b** Staining for IFNγ⁺ NK cells (CD3⁻TCR⁻ Nkp46⁺ or CD3⁻TCR⁻DX5⁺) upon culture with antibodies depicted in panel **a**. **c** Experimental design of adoptive transfer experiments. **d** Representative staining of cellular injection mix and CFSE⁺ splenocytes harvested from recipient mice. Numbers denote percentage of each gated cell population. **e** Relative survival of HLA-C*05⁺ KO cells compared to survival of HLA-C*05⁻KO cells (normalized to WT mice). **f** Representative flow cytometric staining of DX5 and CD49a on uNK cells (CD3⁻CD19⁻TCR⁻CD45⁺Nkp46⁺CD122⁺ cells) isolated from the gd9.5 implantation sites from the mentioned mating crosses. KIR staining is shown on total uNK cells, CD49a⁺, or DX5⁺ uNK subsets. **g** Percentage of KIR⁺ uNK cells in different mating crosses at gd9.5. Crosses are written as female × male. **h** HLA-C*05 mRNA expression on fetal trophoblast cells from gd9.5 or gd10.5 implantation sites.

**i** Representative HLA-C*05 cell surface expression on fetal trophoblast cells isolated from placenta at gd12.5, and cultured with (solid lines) or without stimulation (dashed lines). Purple: HLA-C*05⁺ trophoblast cells, gray: HLA-C*05⁻ trophoblast, i.e., WT cells, filled histogram: fluorescence minus one (FMO) control. Numbers denote MFI. **j** HLA-C cell surface expression plotted as MEF normalized to FMO controls. Mean ± SEM is shown. *n* represents biologically independent animals/samples in each group. **b** *n* = 3 (WT), 10 (HLA-C*05), 4 (KIR2DL1) for α-NKp46 ± α-KIR stim, *n* = 4 (WT), 9 (HLA-C*05), 4 (KIR2DL1) for α-Ly49D ± α-KIR stim. **e** *n* = 9 (WT), 6 (KIR2DL1). **g** *n* = 10 (WT × C*05), 9 (KIR × C*05) and 8 (C*05/KIR × WT). **h** *n* = 4 per group. **j** *n* = 5 per group. *p < 0.05, **p < 0.01, ***p < 0.001, ****p < 0.0001, Two-tailed Mann−Whitney *U*-test. Exact p-values are: **b** *p* = 0.0286 (both comparisons), **e** *p* = 0.0002, **g** *p* < 0.0001, **h** *p* = 0.0286, **j** *p* = 0.0079 (both comparisons). Source Data are provided as a Source Data file.

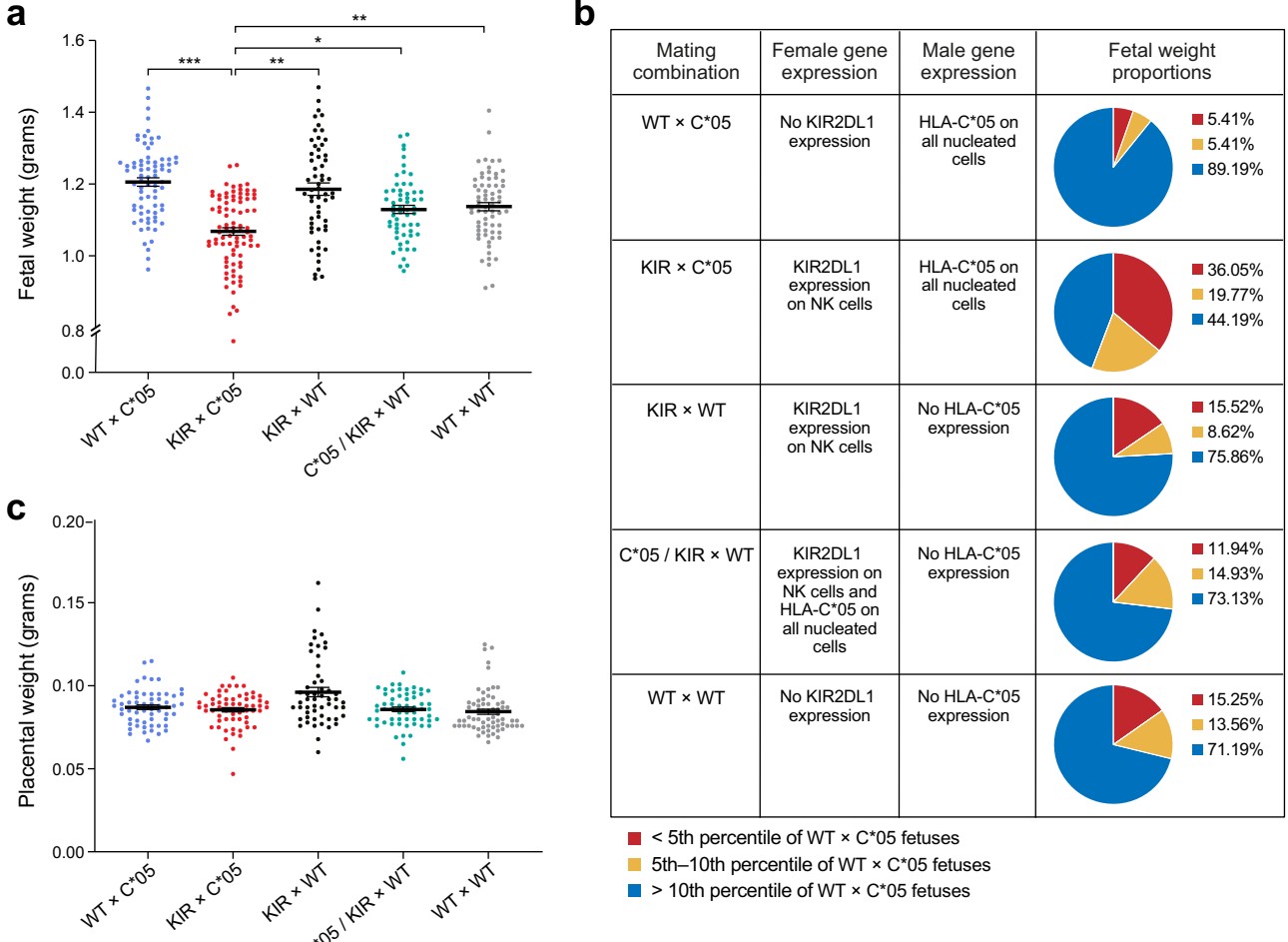

**Fig. 3 | Expression of maternal KIR2DL1 and paternal HLA-C*05 leads to FGR.**
**a** Fetal weight and **c** placental weight determined at gd18.5 of progeny from different mating combinations involving WT, KIR2DL1-expressing and HLA-C*05 transgenic mice. Crosses are written as female × male. **b** Table showing distribution of fetal weight proportions from each mating combination compared to the WT × C*05 control mating. Numbers depict percentages of fetuses whose weight was below the 5th percentile, between 5th and 10th percentile and above the 10th percentile of the WT × C*05 mating controls. Mean ± SEM is shown. *n* represents

biologically independent litters from each group. **a** *n* = 12 litters (WT × C*05), 11 litters (KIR × C*05), 10 litters (C*05/KIR × WT), 10 litters (WT × WT) and 8 litters (KIR × WT). **c** *n* = 10 litters (WT × C*05), 8 litters (KIR × C*05), 10 litters (C*05/KIR × WT), 10 litters (WT × WT) and 8 litters (KIR × WT). *p < 0.05, **p < 0.01, ***p < 0.001, Linear mixed-effects model was used for statistical testing. Exact *p*-values are: **a** *p* = 0.0001 (KIR × C*05 vs. WT × C*05), *p* = 0.0051 (KIR × C*05 vs. KIR × WT), *p* = 0.0198 (KIR × C*05 vs. C*05/KIR × WT), *p* = 0.0043 (KIR × C*05 vs. WT × WT). Source Data are provided as a Source Data file.

## Maternal KIR2DL1 and paternal HLA-C*05 expression leads to changes in uterine spiral arteries during gestation

As compromised uterine circulation may contribute to FGR[17], we looked for changes in maternal uterine spiral arteries that feed the developing fetus at gd10.5 in the KIR × C*05 mating. gd10.5 represented a point when the coiling spiral arteries in the decidua have formed and the uteroplacental circulation has developed[54,55]. Mice

were perfused with an X-ray opaque contrast agent into the uteroplacental circulation and imaged using micro-computed tomography (micro-CT) to preserve the in vivo morphology of the circulatory spaces (Fig. 4a). We manually segmented the spiral arteries in both the KIR × C*05 (FGR) and WT × C*05 (control) mating combinations (Fig. 4b, c) and skeletonized them with homotopic thinning (Supplementary Fig. 3a, b).

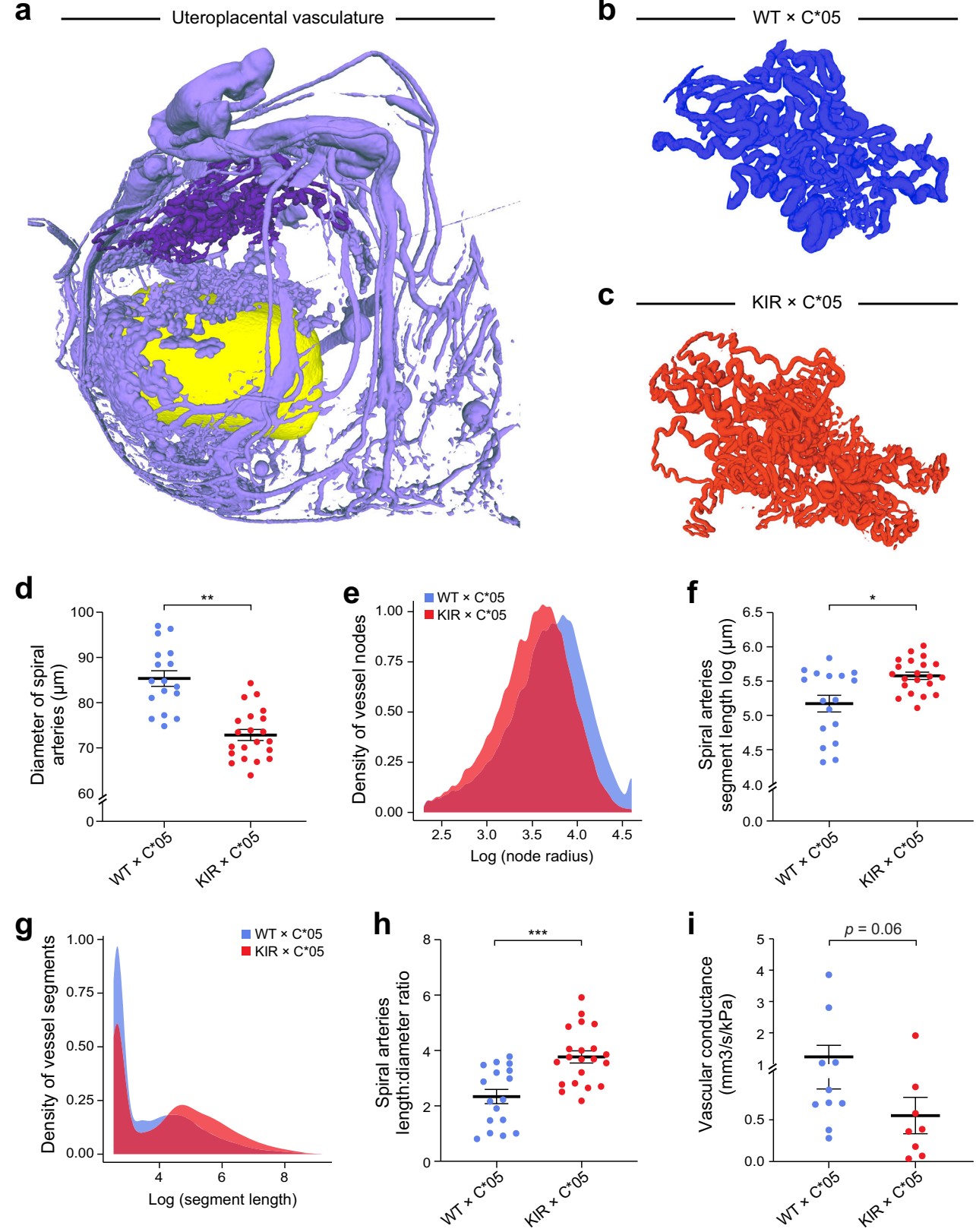

The mean diameter of the spiral arteries was reduced by ~15% in the KIR × C*05 FGR mice vs. WT × C*05 control (p-value < 0.01, linear mixed-effects model, Fig. 4d), also evident by a shift in distribution of radii of the spiral artery vessel nodes (Fig. 4e and Supplementary Fig. 3c). Furthermore, a ~38% increase in the mean spiral artery

segment length was observed in KIR × C*05 FGR mice compared to WT × C*05 controls (Fig. 4f, p-value < 0.05, linear mixed-effects model), alongside a skew in the distribution of the spiral artery segment lengths (Fig. 4g and Supplementary Fig. 3d). KIR × C*05 FGR mice trended towards having fewer spiral artery vessel segments (p = 0.07,

**Fig. 4 | KIR2DL1 and HLA-C*05 interaction alters uterine spiral arteries.**
**a** Representative uteroplacental circulation from WT × C*05 mating at gd10.5 visualized following perfusion with an X-ray contrast agent and imaging by micro-CT. Spiral arteries are shown in dark purple and the yolk sac is shown in yellow. **b**, **c** Representative spiral artery vasculature images from the different mating crosses at gd10.5. **d**, **f** Spiral artery diameter (**d**) or segment length (**f**) is shown. **e** and **g** Radii distribution of spiral artery vessel nodes (**e**) and distribution of segment lengths (**g**) is shown. **h** Ratio of the mean of the spiral artery segment length and mean of the spiral artery diameter of each implantation site is shown. **i** Total network fluid conductance based on the Poiseuille flow model. Mean ± SEM is shown. $n$ represents biologically independent implantation sites from each group. **d**, **f**, **h** $n$ = 17 (WT × C*05) and 21 (KIR × C*05). **i** $n$ = 10 (WT × C*05) and 8 (KIR × C*05). *$p < 0.05$, **$p < 0.01$, ***$p < 0.001$. Linear mixed-effects model was used for statistical testing in **d**, **f**. Exact $p$-values are: **d** $p = 0.0031$, **f** $p = 0.0134$. Two-tailed Mann–Whitney $U$-test used for **h**, **i**. Exact $p$-values are: **h** $p = 0.0003$, **i** $p = 0.0676$. Source Data are provided as a Source Data file.

Mann–Whitney $U$-test), while the total length of spiral artery vasculature and the segment volume remained unaltered (Supplementary Fig. 3e–g).

Owing to the skewed spiral artery length-to-diameter ratio in KIR × C*05 mice (Fig. 4h), we also quantified the total spiral artery network resistance by imposing a pressure gradient between the inlets and outlets and calculating the total outflow. The vascular conductance through the FGR KIR × C*05 spiral artery network tended to be lower than in WT × C*05 mating controls ($p = 0.06$, Mann–Whitney $U$-test) (Fig. 4i), suggesting that there was a higher resistance to blood flow through the spiral artery network in matings that resulted in FGR.

Collectively, these findings demonstrated that it is the interaction between KIR2DL1 on maternal uNK cells and HLA-C*05 on the invading fetal trophoblast cells that changes remodeling of the uterine spiral arteries during gestation, thereby contributing to the development of FGR in KIR × C*05 matings.

## Seven uNK cell subsets at the mid-gestation mouse maternal–fetal interface

To better understand uNK function and diversity at the maternal–fetal interface, we next built a single-cell atlas of uNK cells at mid-gestation (gd9.5), when the frequency of uNK cells peaks at the maternal–fetal interface[24,25], using both full-length scRNA-seq (with SMART-Seq2[56,57], providing deeper coverage per cell, Methods), and 3′ droplet-based scRNA-Seq (profiling large cell numbers, Methods), from the three different mating groups: KIR × C*05 (FGR), WT × C*05 (control 1) and C*05/KIR × WT mice (control 2) (Fig. 5a). For full-length scRNA-seq, we used fluorescence-activated cell sorting (FACS) to sort uNK cells into either trNK (CD49a+) or cNK (DX5+) cells (Fig. 5b, Supplementary Fig. 4a and Supplementary Fig. 5a–c), whereas for droplet-based scRNA-seq, we profiled total uNK cells (Fig. 5d, Supplementary Fig. 4b, and Supplementary Fig. 5d–f). cNK and trNK cells formed two distinct subsets in a low dimensionality embedding (Fig. 5b), and differed in signatures of differentially expressed genes (Fig. 5b, c and Supplementary Data 1). cNKs expressed cellular migration and homing markers (e.g., *Ccl5*, *Sell*) and had high *Gzma* expression, while trNKs expressed cell–cell and cell–matrix interaction markers (e.g., *Adam8*, *Spp1*) and had high *Gzmc* expression (Fig. 5c). trNKs had more genes detected per cell (median number of genes per cell—trNKs = 4258 and cNKs = 2165; $p$-value = $6.96 \times 10^{-264}$, Mann–Whitney $U$-test) and a larger size by FACS than cNKs, consistent with higher transcriptional activity (Supplementary Fig. 5a); uNK cell subset size differences have been demonstrated microscopically[58]. Scoring the droplet-based profiles with the full-length-based cNK and trNK signatures similarly distinguished the two populations in the low dimensionality embedding (Fig. 5d). As reported previously[59], cNK cells represented a smaller proportion of total profiled uNK cells (Fig. 5d and Supplementary Fig. 1n). We used unsupervised graph-based clustering (with the Louvain algorithm[60]) to identify six NK subsets in the full-length scRNA-seq data (Fig. 5b) and eight NK subsets in the droplet-based scRNA-seq data (Fig. 5d). After comparing and evaluating subsets across both datasets, we annotated seven distinct NK cell subsets that were concordant between the two datasets (Fig. 5e and Supplementary Fig. 6a, b). Owing to the deeper gene coverage, we prioritized the subset-specific differentially expressed genes from full-length scRNA-seq to

annotate six of the NK subtypes found in the full-length data (Fig. 5f): (1 and 2) two cNK subtypes (cNK-1 and cNK-2) that shared several cNK marker genes, however, only cNK-2 had NF-kB inhibitors (*Nfkbiz*, *Nfkbid*, *Nkfkia*), (3) a trNK cell subset (trNK-1) enriched for cytokine receptor signaling genes and marked by *Cd7*, *Cxcr6* and *Pdcd1*; (4) *Gzmd/e/g/f*-high trNKs (trNK-2) enriched for several central and protein metabolic pathways (e.g., glycolysis, $p = 3.1 \times 10^{-5}$, Gene set enrichment analysis—Fisher's exact test), (5) proliferating trNKs (trNK-3 or trNK-3a/3b) expressing *Birc5* and *Ccna2*, and having a high gene proliferation signature score (Supplementary Fig. 6c and Supplementary Data 2), and (6) a subset including both cNK and trNK cells (mix-1), defined by expression of *Cd27*, *Cd7* and *Emb* (the gene signature for this mix-1 subset scores higher on cNKs than trNKs, Supplementary Fig. 5g–h). Of the above six subsets, only cNK-2 did not form a distinct subset in the droplet-based data, however, scoring its signature highlights a distinguishable population within the cNK-1 cluster of the droplet-based data (Supplementary Fig. 6a). The seventh NK subset, *Gzmd/e/g/f*-high trNKs expressing *Pdcd1* (trNK-4), was a distinct subset only in the droplet-based data. Scoring its signature identified it within the trNK-2 subset in the full-length data, but these cells largely overlapped with the full-length and droplet-based trNK-2 cluster and signature, and it is possible that this subset was over-split in the droplet-based data. The eighth subset found in the droplet-based data (trNK-5) was less well defined and not included as a major uNK cell subset in our final census (Fig. 5b, d, e, f, Supplementary Fig. 6b, and Supplementary Data 3).

Overall, cells from KIR × C*05 (FGR), WT × C*05 (control 1) and C*05/KIR × WT (control 2) matings had similar distributions across most of the subsets identified by clustering, suggesting that genotype did not have a dramatic effect on the overall NK cell subset categories (Supplementary Fig. 5i–n), with the exception of the *Gzmd/e/g/f*-high, *Pdcd1*-high trNK-4 cluster (in droplet-based data), which was enriched for cells from the KIR × C*05 (FGR) genotype ($p < 0.05$, Dirichlet-multinomial regression[61], Supplementary Fig. 5n). Subset trNK-4, however, was not distinct from trNK-2 in the full-length data and shared many marker genes with trNK-2 in both datasets.

Collectively, these results highlighted the diversity of uNK cells at mid-gestation and identified seven transcriptionally distinct subsets of cNK and trNK cells present at the maternal–fetal interface in control and FGR mice.

## Increased lymphocyte activation and decreased extracellular matrix (ECM) and tissue remodeling programs in uNK cell subsets in FGR

Given the continuum of uNK cell subsets and the limited changes in proportion of these subsets between genotypes, we hypothesized that FGR may have more nuanced changes in cell intrinsic gene programs within uNK subsets. These may not be recovered by the clustering and pathway analysis approach above, either because of lack of discrete functions, or in cases when the same process is active in cells of different types[62]. To uncover those, we used an approach involving topic modeling with latent Dirichlet allocations (LDA)[62–64] to learn 16 gene programs ("topics") across the droplet-based scRNA-seq profiles (Supplementary Fig. 6d, Methods). This approach assigns each gene and cell a weight in every topic, indicating the importance of the gene to the topic and of the topic to the cell, with the top scoring genes

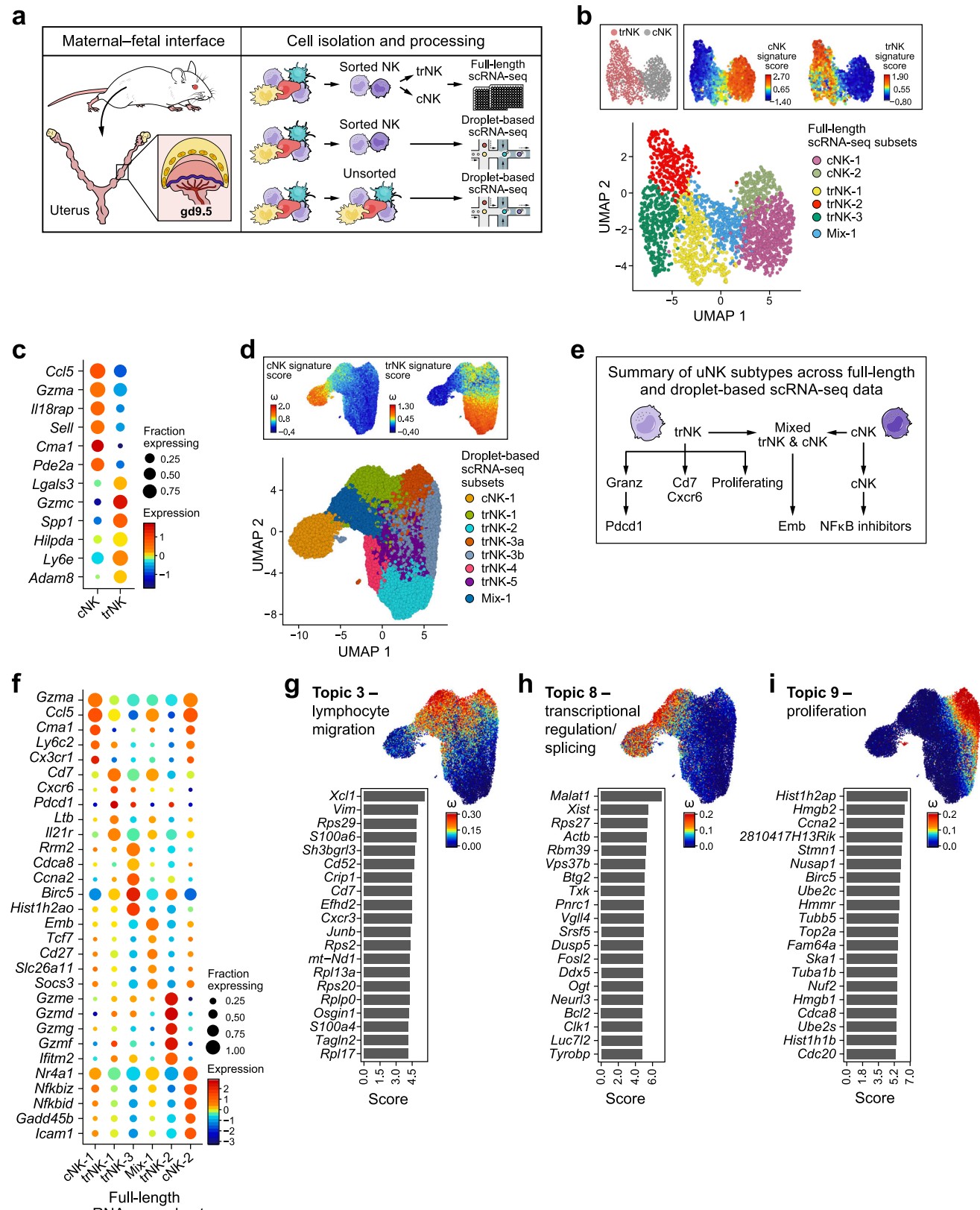

defining the gene program for that topic. Cells and genes can be highly weighted for more than one topic.

Overall, topics ranged from general biological processes to regulation of NK cell function, and included cNK/trNK-specific topics and those spanning cells from both subtypes (Supplementary Fig. 7 and Supplementary Data 4). For example, Topic 3, Topic 8 and Topic 9

represented programs involved in lymphocyte migration, transcriptional regulation and splicing, and proliferation, respectively, and spanned across different uNK subsets (Fig. 5g–i and Supplementary Figs. 6 and 7).

Looking more specifically at uNK cell topics altered in FGR, the cell weights for topic 6 (lymphocyte activation) and topic 16 (ECM and

**Fig. 5 | Heterogeneity of NK cells at the mouse maternal–fetal interface.**
**a** Overview of study design. **b** UMAP embedding of full-length scRNA-seq on sorted NK cells colored by subtype as cNK or trNK (determined by FACS sorting and verified using the cNK and trNK signature score) or by Louvain cluster label. **c** Dot plot showing top six cNK and trNK marker genes. **d** UMAP embedding of NK cells profiled by droplet-based scRNA-seq and colored by Louvain cluster or the cNK and trNK signature score. **e** Summary of NK subtypes identified in sorted uNK cells derived from both full-length and droplet-based scRNA-seq data. **f** Dot plot of top five marker genes for each cluster shown in **b**. **g**–**i** Left: top twenty genes driving the topic; the score has been scaled to improve visualization. Right: UMAP embedding

of droplet-based scRNA-seq uNK data (same as in panel **d**) colored by the weight of each cell in the topic. Topic 3 (**g**), Topic 8 (**h**), Topic 9 (**i**). In all plots, cells from all three genotypes (WT × C*05, KIR × C*05, C*05/KIR × WT) are included. For full-length scRNA-seq, $n = 5$ mice for all three mating groups, $k = 2176$ cells (trNK = 1091 and cNK = 1085). For droplet-based scRNA-seq, $n = 4$ mice (WT × C*05) and 3 mice (KIR × C*05 and C*05/KIR × WT), $k = 30,147$ uNK cells. Each mouse represents cells pooled from the maternal–fetal interface of multiple implantation sites within the same litter. In dot plots, the size of the dot represents the fraction of cells with nonzero expression of each gene, and the color of the dot represents the average nonzero gene expression. Source Data are provided as a Source Data file.

tissue remodeling) were significantly and robustly different in cells from KIR × C*05 FGR mating vs. control matings (Fig. 6a–c, Methods). The lymphocyte activation program (Topic 6) is induced in FGR ($p$-value = $5.06*10^{-9}$, KS-statistic = $-0.049$ for FGR vs. CTR1; $p$-value = $4.09*10^{-14}$, KS-statistic = $-0.073$ for FGR vs. CTR2; KS-test), whereas the ECM and tissue remodeling program (Topic 16) is repressed in FGR ($p$-value = $3.3*10^{-3}$, KS-statistic = $0.028$ for FGR vs. CTR1; $p$-value = $3.34*10^{-5}$, KS-statistic = $0.043$ for FGR vs. CTR2; KS-test).

The lymphocyte activation program induced in FGR (Topic 6) had the highest cell weights in the region of the droplet-based cNK-1 subset expressing NF-kB inhibitors (aligned with full-length cNK-2 subset) and the mixed cNK/trNK subset (mix-1). This gene program showed enrichment for NF-kB signaling pathway, chemokine signaling pathway, cytokine-cytokine receptor interaction genes and ribosomal genes ($p = 0.0008, 0.0131, 0.048, 1.37*10^{-32}$, respectively; Gene set enrichment analysis–Fisher's exact test). Top genes in this topic included *Xcl1*, *Ccl1* and *Ccl4*, which directly impact trophoblast and other cells in the decidua such as macrophages and dendritic cells, specifically influencing their recruitment and migration at the maternal–fetal interface[65–67]. In situ hybridization on mid-gestation uterine tissue also confirmed increased expression of *Ccl1* in KIR ×C*05 FGR mice vs. controls (Fig. 6d). Another topic gene–*Cd7* is involved in costimulatory triggering and can augment function of adhesion molecules on NK cells[68]. *Gadd45b* was a NF-kB-regulated topic gene involved in regulation of cell growth and apoptosis, and whose transcript levels are increased upon response to physiological and environmental stresses[69,70]. Presence of several ribosomal genes in this topic (in the absence of active proliferation) was also likely an indication of cellular stress[71].

The ECM and tissue remodeling program repressed in FGR (Topic 16) has cell weights highest in the trNK-2 subtype, which had high expression of *Gzmd/e/g/f*. Top program genes included *Gzmd*, *Gzmg* and *Gzme*, but not *Gzmf*. These granzymes have been shown to be expressed by uNK cells in pregnancy and are upregulated by IL-12 and IL-15 in uterine decidual cells (the modified uterine endometrium formed during pregnancy). Unlike *Gzma* and *Gzmb*, these granzymes (*d*, *g*, and *e*) are proposed to be non-cytotoxic and have functions in ECM and tissue remodeling, as well as in parturition[72–74]. Decreased expression of *Gzmd/e/g* was also confirmed by in situ hybridization on mid-gestation uterine tissue in KIR × C*05 FGR mice vs. controls (Fig. 6e). Downregulation of this expression program involving *Gzmd*, *e* and *g* genes in FGR matings provides insights into the function of these granzymes in regulating placentation and consequently, fetal growth. Other topic program genes are also involved in tissue remodeling (*Ctsg*), protection against oxidative damage (*Mt1* and *Mt2*)[75,76] and regulation of maternal–fetal tolerance (*Havcr2* or *Tim-3*, with Tim-3+ NK cells being associated with an immunosuppressive phenotype)[77–79]– implicating these uNK cell functions in regulating fetal growth.

In addition to topic modeling, we also tested for genes that are differentially expressed in cells from the FGR phenotype within each specific NK cell subset/cluster identified using the unsupervised Louvain clustering approach (differentially expressed genes were calculated using pseudobulk differential expression analysis, Methods,

Fig. 6a, f and Supplementary Data 5). Some cell subset-specific differentially expressed genes in addition to those highlighted by topic modeling include *Fos* (cNK-1), *Acp5* (trNK-1), *Spp1* (trNK-3), *Gsto1* (trNK-1), *Map3k1* (trNK-1, trNK-3), *Akt3* (trNK-2), and *Il10rb* (trNK-2) (Fig. 6f). These genes were involved in modulation of NK cell function, stress response and signaling pathways in the placenta. For example, *Fos* encodes for one of the proteins in the AP-1 transcription complex and modulates NK cell function[80]; *Acp5* encodes an enzyme (TRAP) that regulates the activity of Osteopontin (encoded by *Spp1*–also a differentially expressed gene), involved in regulation of fetal growth and recurrent spontaneous abortion[81,82]; *Gsto1* is a stress response gene, associated with redox homeostasis[83]; *Akt3* is a protein kinase family member, which can also promote induction of reactive oxygen species[84]; *Il10rb* is a cell surface receptor for multiple cytokines and can stimulate activation of the JAK/STAT signaling pathway[85], and *Map3k1* integrates multiple signaling pathways and can also crosstalk with Wnt signaling–a key signaling pathway in decidualisation–which is critical for remodeling the uterine endometrium, regulation of placental growth and trophoblast invasion[86,87], which is likely to be impacted in FGR.

Together, our topic modeling and cluster-based differential gene expression approaches show that in FGR there is an increase in programs related to regulation of cell growth, lymphocyte recruitment, apoptosis and NF-kB signaling, and decrease in programs for ECM and tissue remodeling and protection against oxidative stress alongside genes that impact key signaling pathways in the placenta. Considering that these expression changes were detected at a gestational stage much earlier than the onset of FGR, they likely represent causative mechanisms for FGR and provide insights into programs affected in uNK cells as a result of the interaction between KIR2DL1 and HLA-C*05.

## FGR-associated transcriptional changes across diverse cell types at the maternal–fetal interface
To understand the function of NK cell gene programs in the broader context of the cellular ecosystem, we analyzed all cell types at the maternal–fetal interface at mid-gestation using droplet-based scRNA-seq, and identified transcriptional differences in FGR by cell type (Fig. 7a and Supplementary Fig. 8a–c).

Unsupervised clustering and post-hoc annotation highlighted 20 cell subsets (Methods), spanning immune, stromal, decidual, trophoblast and vascular smooth muscle (vSMCs) cells (Fig. 7b and Supplementary Data 6), similar to findings from early human pregnancies[88]. There were nine immune cell subsets: trNKs, mixed cNKs and trNKs, (Supplementary Fig. 8d), T cells, B cells, ILC2/3 s, dendritic cells (*Cd209a+*), two macrophage subsets differing in complement (*C1qa*, *C1qb*) and MHC II (*H2-Ab1*, *H2-Eb1*) expression, and a myeloid subset (expressing *Retnlg*, *Slpi* and *S100a9*)[89]. There were seven stromal cell subsets: one of endometrial stromal fibroblasts (*Col3a1*, *Col1a1*, *Col1a2*, *Dcn*, and *Lum*), and six stromal subsets at different stages of cellular differentiation or decidualisation as reflected by the level of Wnt signaling molecules and modulators (*Wnt4*, *Wnt5a*, *Wnt6*, *Sfrp1*)[90,91], TGF-β superfamily morphogen (*Bmp2*)[90,91], angiogenic factors (*Angpt2*, *Angpt4*) and steroid biosynthesis genes (*Cyp11a*). One non-canonical subset was decidual-like, which likely represented cells that support

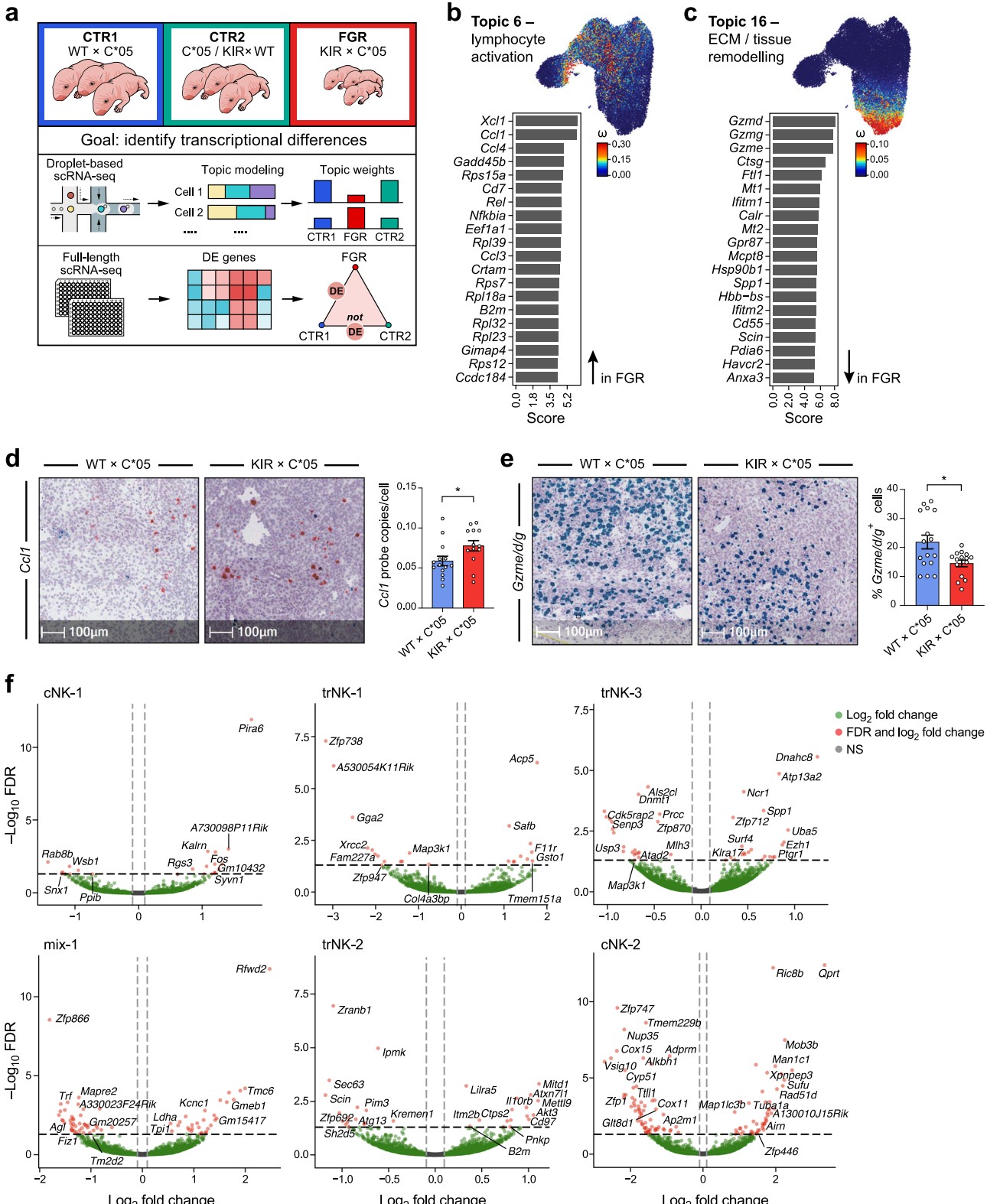

decidualisation—this subset expressed the decidual prolactin-related protein *Prl8a2*, *Cyrab* (important in embryo implantation and decidualization) and Uteroglobin-encoding[92] *Scgb1a1* (Fig. 7b and Supplementary Data 6).

There were three trophoblast subsets: Trophoblast 1 cells expressed *H19*, which influences trophoblast cell migration and invasion[93], and other trophoblast genes, including *Bex1*, *Krt8* and *Mest*, and included cells expressing endothelial markers (e.g., *Pecam1*, *Cd34*,

*Tie1*, *Egfl7*). Trophoblast 2 cells expressed *Hsd11b2* (an enzyme produced by trophoblast cells[94]) and the prolactin receptor *Prlr*, which influences trophoblast invasiveness and migration[95]. Trophoblast 3 cells expressed keratins (*Krt7*, *Krt8*, *Krt18*, *Prap1*, expressed by the uterine luminal epithelium[96]) and the early trophoblast marker *Wfdc2*[97]. Finally, VSMCs expressed actin-binding and contractile genes such as *Acta2*, *Myl9*, *Cnn1* and *Tagln* (Fig. 7b and Supplementary Data 6).

**Fig. 6 | FGR phenotype characterization in NK cells. a** Overview of FGR phenotype characterization by topic modeling and differential gene expression analysis. **b, c** Topic modeling of droplet-based scRNA-seq uNK cell data from the mouse maternal–fetal interface. The cell topic weight distributions for the displayed topics Topic 6 (**b**), and Topic 16 (**c**) were significantly different between KIR × C*05 vs. control mating groups. Left: top 20 genes driving the topic; the score has been scaled to improve visualization. Right: UMAP embedding colored by the cell weight for that topic—bright red indicates the cell is high in the topic and dark blue indicates the cell is low in the topic. **d, e** Re-validation of *Ccl1* (**d**) and *Gzme/d/g* (**e**) from the topics using in situ hybridization on gd9.5 implantation sites. Representative probe staining on sections from the different mating crosses is shown.

Quantification depicted as average probe copies per cell or percentage of positive cells within a stained section for the respective probes is shown as mean ± SEM. **f** Volcano plots showing differentially expressed genes in the FGR KIR × C*05 mating combination. Differentially expressed genes were calculated on a per cluster basis, and are stratified as such in the panel. **b, c** $n$ = 4 mice (WT × C*05) and 3 mice (KIR × C*05 and C*05/KIR × WT), $k$ = 30,147 uNK cells. **d, e** $n$ = 14 (WT × C*05) and 13 (KIR × C*05) independent sections for *Ccl1* and $n$ = 16 (WT × C*05) and 15 (KIR × C*05) independent sections for *Gzmd/e/g*, *$p$ < 0.05, Two-tailed Mann–Whitney $U$-test. Exact $p$-values are: **d** $p$ = 0.0332, **e** $p$ = 0.0448. Source Data are provided as a Source Data file.

Cell subset composition was similar across different mice and genotypes (Supplementary Fig. 8e–g), but we identified FGR-associated gene expression changes in cell intrinsic expression in multiple cell types, either shared across cell types, or specific to a subset (Pseudobulk differential expression analysis, Fig. 7c and Supplementary Data 7). This suggests a cascading and amplifying effect where the initial interaction between KIR2DL1 on maternal uNK cells and HLA-C*05 on fetal trophoblast cells had major and broad downstream effects on multiple cell types present at the maternal–fetal interface. For example, *Anxa2* (a member of the Annexin family with functions in cell migration, invasion and angiogenesis[98]) was downregulated in cells from FGR mice compared to controls in different cell types (Fig. 7c), which we validated in situ (Supplementary Fig. 8h). Surprisingly, there were cell type-specific differentially expressed genes in many non-NK cell subsets in the FGR mating group. These included stromal cell genes had functions in angiogenesis (*Rnf213, Slit3*[99,100]), regulation of fetal growth (*Igf1, Igfbp2, Igfbp3, Igfbp4*[101]), invasion and migration (*Usp25, Prlr, Aspn, Spon2*[95,102–104]), decidualisation and Wnt signaling (*Sfrp4, Rnf213, Wnt4*[105,106])–disease functions, which were also highlighted by pathway analysis (Fig. 7c and Supplementary Fig. 8i). Macrophages and dendritic cells showed changes in genes including *Osm*, involved in invasion under hypoxic conditions[107], and transcription factor *E2f1* that affects dendritic cell maturation[108]. Differentially expressed genes in FGR in macrophage cell subsets were enriched for dendritic cell maturation, antigen presentation or inflammation (FDR = 0.007, 0.02, 0.0001, Ingenuity pathway analysis) (Supplementary Data 8). Analysis of differentially expressed genes and pathway analysis demonstrated that trophoblasts and vSMCs showed FGR-associated changes in genes involved in invasion (*Klf5*[109]), regulation of angiogenesis or ECM remodeling (*Klf5, Cyr61, Col4a2, Bmper, P4ha1*[110–114]) and stress response (*Sparc*[115]) (Fig. 7c and Supplementary Fig. 8j). Some of these genes have previously been implicated in the pathogenesis of pre-eclampsia (*Sfrp4, Wnt4, Sparc, Rgs2, Cyr61, E2f1*[116–122]) and small-for-gestational-age pregnancies (*Dcn, Igf1, Igfbp2, Igfbp3, Igfbp4*), however, several of the other genes and pathways highlighted here represent new candidates that shed light on disease-relevant functions changed as a result of the KIR2DL1-HLA-C*05 interaction, which have potential for therapeutic intervention.

Collectively, these analyses show that the initial interaction between KIR2DL1 and HLA-C*05 leads to early changes in transcriptional expression and modulation of placental pathways in multiple cell types at the maternal–fetal interface in FGR.

### Maternal KIR2DL1 and paternal HLA-C*05 lead to change in cell–cell interaction networks in FGR

Finally, we analyzed our data to identify changes in putative cell–cell interactions in FGR vs. control mice. We identified putative interactions between cell subsets (including within the same cell type) based on expressed receptor–ligand (R–L) pairs (Fig. 7a, Methods), scored significant interactions with CellPhoneDB in WT × C*05 control and KIR × C*05 FGR groups separately (Fig. 7d, e and Supplementary

Data 9), and highlighted the interactions, which were increased (red) or decreased (blue) between any pair of cell subsets in the FGR network (Fig. 7e and Supplementary Data 9).

At baseline, the highest number of significant interactions occurred between stromal cells, fibroblasts, trophoblasts, and vSMCs, and both cNKs and trNKs had significant interactions with most other cell types (Fig. 7d). There was a total of 399 cell subset pairs with significant interactions in at least one condition. 38 of these pairs had at least 20% more significant R–L interactions in FGR than in WT (some as many as 50% more). Conversely, only 9 of the cell subset pairs had fewer significant R–L interactions in FGR (Fig. 7e). As more cell subset pairs had an increase rather than a decrease in the number of significant interactions in FGR, we hypothesize that there is an overall increase in predicted intercellular communication at the maternal–fetal interface in FGR.

To identify R–L interactions that may be important in driving disease at the maternal–fetal interface in the early stages of gestation, we focused on putative R–L interactions between NK cells and the other cell types that were detected only in FGR or those that were detected only in controls (selected examples in Fig. 7f). For example, the interaction between *Ccl5* on NK cells and *Ccr1* on macrophages is only significant in the FGR group; *Ccl5-Ccr1* interaction is involved in decidual immune cell recruitment[123]–in line with increased intercellular communication at the maternal–fetal interface observed in the FGR group. There is a significant interaction between *TgfβR1* on NK cells and *Tfgβ1* on macrophages and dendritic cells in the FGR group but not in the control group. *Tfgβ1* has been shown to suppress activation of decidual NK cell subsets and blockage of *Tfgβ1* improved decidual NK cell-mediated angiogenesis[124], thus implicating this pathway in the pathology of FGR. Similarly, considering that the Notch pathway is a key pathway regulating angiogenesis[125], there was interaction between *Notch1* in NK cells and *Dlk1* in trophoblast cells or SMCs in the FGR mating but not the control group. Furthermore, TGFβ is involved in the control of trophoblast invasion, proliferation and migration[126], and there was a significant interaction between *TgfβR1* on NK cells and *Tfgβ2* on the stromal, trophoblast and SMC subtypes in FGR. We also noted an interaction between *Mif* on NK cells and *CD74* on stromal cell subsets in the WT, but not the FGR group; *Mif* has been proposed to promote stromal cell survival, migration and invasion of trophoblast cells–functions which were evidently affected in FGR[127].

Overall, these results highlight potential functions for specific R–L interactions between uNK cells and other cell types at the maternal–fetal interface in early gestation, which have not been described before in the context of FGR. Targeted interference of these disease-specific R–L interactions could create new opportunities for therapeutic intervention.

## Discussion

FGR affects 5–10% of all pregnancies, and is a leading cause for perinatal morbidity, mortality and long-term health issues for the child. However, clinical practice has remained unchanged for decades, and treatment as well as prevention strategies are lacking due to insufficient insight into the pathogenesis of FGR. While it is known that

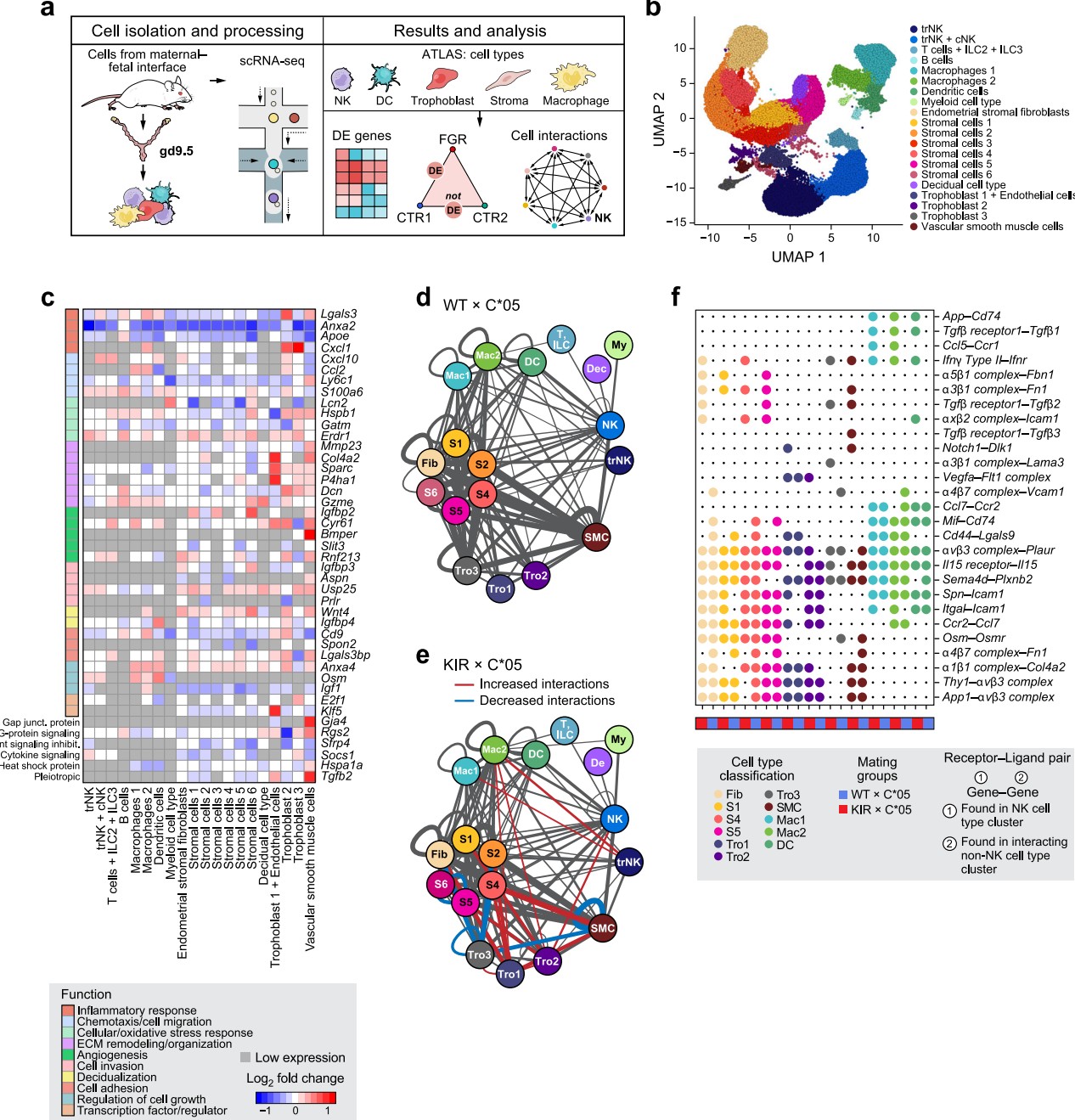

**Fig. 7 | The maternal–fetal interface atlas and cell–cell interactions in FGR.**
**a** Overview of scRNA-seq analysis of unsorted cells from the maternal–fetal interface. **b** UMAP embedding of unsorted cells colored by cell type classification.
**c** Heatmap of select differentially expressed genes (calculated on a per cell type basis) between KIR × C*05 and WT × C*05 mating groups. Every gene in the heatmap was differentially expressed in at least one cell type. Color of square indicates log of the average expression fold change between KIR ×C*05 and WT × C*05. Bright red: higher average expression in KIR × C*05; Dark blue: lower average expression in KIR × C*05; White: no difference; gray: gene expressed in <10% in both conditions within a cell subset. Colored boxes to the left indicate top functional annotation of each gene. **d**–**e** Cell–cell interaction analysis by CellPhoneDB. Cell–cell interaction network for WT × C*05 (**d**), and KIR × C*05 (**e**) cells. Color of circle corresponds to the cell type classification in panel **b**. Line weight represents

number of significant ligand–receptor interactions found between the cell type nodes. Interactions increased or decreased in KIR × C*05 group compared to WT × C*05 group are shown in red and blue, respectively. **f** Select ligand–receptor pairs unique in either KIR × C*05 or WT × C*05 groups in one of the NK cell types. The first gene listed in the label was found in one of the NK clusters, and the second gene was found in an interacting non-NK cell type cluster. Large circle indicates that R–L pair is found between an NK cluster and the indicated cell type, small circle indicates absence of the R–L pair within the interacting cell types; cell types are colored as in **b**. **b** *n* = 3 mice each from WT × C*05, KIR × C*05, and C*05/KIR × WT, *k* = 42,869 cells. Each mouse represents cells pooled from the maternal–fetal interface of multiple implantation sites within the same litter. Source Data are provided as a Source Data file.

certain classes of genes predispose to FGR, the exact identification of these genes as well as their functional impact on disease pathogenesis have remained unknown. Employing a functional genetics approach, we define how two immune genes, i.e., *KIR2DL1*, expressed on maternal

uNK cells, and paternally derived *HLA-C*05*, expressed on fetal trophoblast cells, lead to FGR in a humanized mouse model. Importantly, we demonstrate how a relatively discreet interaction between these two immune genes cascades into cellular crosstalk and transcriptional

expression changes in a myriad of cells at the maternal–fetal interface, thus providing an understanding of FGR as a complex multicellular disease. These findings provide mechanistic insight into disease and highlight disease functions and gene candidates, which have the potential to be modulated for therapeutic intervention.

Previous studies have suggested that uterine arterial remodeling is mediated by uNK cells, largely through secretion of IFNγ, without involvement from the trophoblasts[27]. However, our data demonstrates that this mechanism is more complex, and involves interaction between the maternal uNK and fetal trophoblast cells, mediated through KIR-HLA, resulting in pathogenic uterine remodeling. While uNK cells might be able to independently mediate some uterine remodeling in early gestation, our data highlights that it is the inter-action between the KIR on uNK cells and HLA on trophoblast cells that fine tunes the critical arterial remodeling, which potentiates FGR. In line with the genetic studies[30], we only observed FGR when *HLA-C\*05* was inherited paternally. We did not observe significant FGR in the C\*05/KIR × WT mating combination, where the group 2 *HLA-C* gene in the fetus was maternally inherited, and the mother and fetus had the same level or number of copies of group 2 *HLA-C* genes. Our data mimics the paternal effect from the human genetic association studies; this not only adds credibility to our humanized model, but also underscores the disease relevance of the specific *KIR* and *HLA* alleles highlighted in our study.

We highlight different cellular functionalities and gene programs that are important in FGR at a critical timepoint in early gestation from the pregnant mouse uterus—something which cannot be addressed in studies employing human placental samples[88,128]. We observe that subtypes of both the cNK and trNK cells are involved in biological processes altered or deficient in FGR. The uNK functions implicated in disease etiology of FGR include lymphocyte recruitment and migra-tion, cellular adhesion, ECM tissue remodeling, protection against oxidative damage, and stress response, among other key signaling pathways in the placenta. Importantly, these alterations in gene pro-grams and functions are identified at an early gestational timepoint that coincides with the onset of placentation and observed changes in the uterine spiral artery vasculature, while the disease is detected later in gestation. Therefore, it is not plausible that the reported changes in gene expression and biological programs are a consequence or a compensation for FGR, instead, they likely represent causative mechanisms as a result of the KIR2DL1:HLA-C\*05 interaction that contribute to the development of FGR.

To capture disease-related changes in other cell types at the maternal–fetal interface, we profiled all cells at mid-gestation. We identified a range of immune, decidual, stromal and trophoblast cell types, which corroborated with the single-cell atlas from healthy human pregnancies[88], demonstrating the relevance of this huma-nized transgenic model. Despite the KIR2DL1:HLA-C\*05 interaction at the maternal–fetal interface being driven by NK and trophoblast cell types, substantial transcriptional changes in gene expression were observed in FGR in several other cell types at the maternal–fetal interface—suggestive of a cascading effect down-stream of the initial targeted cell types. These changes impacted key functions and pathways such as decidualization, Wnt signaling, cellular invasion, among others, thereby implicating these in the etiology of FGR. Our data indicate that this is a result of the cellular crosstalk between uNK cells and the cells in the microenvironment through specific R–L interaction pairs present exclusively in the FGR mice or control mice, and these disease-specific R–L interactions could be amenable to therapeutic intervention. Our work identifies multiple cell types and pathways implicated in the complex pathogenesis of FGR, opening up a wide range of treatment opportunities including repurposing the currently available ther-apeutic interventions.

The gene expression changes in cells at the maternal–fetal inter-face in FGR were moderate in magnitude but spanned many genes and cell subsets. This combined effect is likely to be further amplified due to the sensitivity to changes to placentation and the fetus at this early and critical stage of gestation. In line with this, healthy pregnancy represents a fine-tuned and highly calibrated system and a disturbance in that balance leads to a multitude of downstream effects, ultimately manifesting in disease. We saw striking similarities in genes and pathways that were differentially expressed in FGR with those that are involved in tumor progression and metastasis. For example, processes such as cellular invasion, migration, angiogenesis, cellular proliferation and differentiation are important in placentation, but are also involved in malignancy. Unlike the uncontrollable growth that occurs in cancer, these processes, however, are normally tightly controlled in develop-ment of the placenta[129,130].

FGR is difficult to predict and it has been found that growth restricted fetuses (specifically those >10th percentile in size) can remain undiagnosed despite not meeting their growth potential and having an increased risk for adverse perinatal outcome[19,23,131]. There-fore, identifying specific risk genes, which can be used to screen par-ents and identify infants with increased risk of neonatal complications, is an important step towards reducing the burden of FGR. Investigation of the reasons underlying adverse perinatal outcome in FGR fetuses would be an important point for future studies. Our work paves the way towards a comprehensive diagnostic screening process to identify pregnancies at greater risk of developing FGR, which would benefit from better monitoring and early clinical intervention in due time, thereby offsetting the long-term risk of serious health implications. A central unanswered question in the field has been how *KIR* and *HLA-C* genetic findings translate into functional mechanisms that lead to an increased risk of developing disease. Our study sheds light on this by first recapitulating the risk gene combination, and then demonstrating the function of different cell types/subsets and functions implicated in the complex pathogenesis of FGR that can now form the basis of new therapeutic interventions, which can improve clinical outcomes for both the mother and the baby.

## Methods

### Animals

All animal experiments were approved by the local Ethical Review Committee at the University of Oxford, and performed under license from the UK home office (project license numbers 30/3386 and P0A53015F) in accordance with the Animals (Scientific Procedures) Act, 1986. Mice (species: *Mus Musculus*) were maintained in a pathogen-free facility in individually ventilated cages in an ambient temperature- and humidity-controlled room with a 12 h light/12 h dark cycle under standard housing conditions with continuous access to food and water. The HLA-C\*05 and KIR2DL1 transgenic mice were generated on the C57BL/6J × 129 background as described below. The mice were further backcrossed onto C57BL/6J for 4–5 generations. Mice at the same backcrossing generation were used as transgene-negative littermates or controls within each experiment. The NCR1-iCre mice were on the C57BL/6N background and were a gift from Veronika Sexl[46]. H2-KbH2-Db double-knockout mice were on the C57BL/6 background and were a gift of Petter Höglund. Adult male and female mice within the age of 8–36 weeks were used for character-ization and functional experiments. Adult female mice within the age of 8–30 weeks were used for timed-mating experiments. None of the mice had any noticeable health abnormalities. For timed-mating experiments, male and female mice were mated in the afternoon of one day and checked for plugs and separated on the morning of the next day. Detection of the plug in the female mice was considered as gd0.5. In addition, female mice were weighed prior to being set-up in timed-mating experiments, and on every alternate day from gd7.5

onwards in order to confirm pregnancy. These refinements avoided wastage of any pregnant transgenic mice. The exact stage of gestation, i.e., gd9.5, 10.5, 12.5, or 18.5 experiment is specified for every experiment.

## Generation of transgenic mice

The HLA-C*05 and KIR2DL1 transgenic mice were generated using targeted insertion into the *ROSA26* locus. For generation of the HLA-C*05 transgenic mice, to facilitate interaction of HLA-C with murine CD8 (and hence the T-cell receptor), the α3 domain of HLA-C*05 was replaced with its murine counterpart from H2-K$^b$, along with the adjacent transmembrane and cytoplasmic domains. To allow for this, the *HLA-C*05* construct was made by amplifying ~2.04 kb genomic fragment of *HLA-C*05:01:01:01*, which contained 776 bp of the *HLA-C* 5′UTR and exons 1–3 up to a midpoint in intron 3, which was fused to a ~3.58 kb fragment of the genomic *H-2K$^b$* gene, beginning at a midpoint in intron 3 and containing exons 4–8 and the *H-2K$^b$* 3′UTR. The genomic DNA construct was first assembled in the pTZ18U vector (Sigma-Aldrich) prior to subcloning into vector pCB92[132], downstream of a promoterless neomycin selection cassette and a murine H19 insulator, creating plasmid CB92-C*05 in which the whole transgene array is flanked by PhiC31 attB sites.

For generation of the KIR2DL1 transgenic mice, the coding DNA sequence of *KIR2DL1*0030201* along with a Kozak consensus sequence (GCCACC) immediately upstream of the ATG start codon, was amplified and cloned into the pEx-CAG-stop-bpA vector (a gift from Ralf Kühn), between the loxP flanked STOP cassette and the polyadenylation (pA) signal. The presence of the STOP cassette (a puromycin resistance coding region followed by triple pA signals) served as the transcriptional STOP signal for transgene expression. The KIR2DL1 protein is expressed from the CAG promoter upon the Cre mediated excision of the loxP flanked stop cassette. This vector also included the promoterless neomycin selection cassette and PhiC31 attB sites required for PhiC31 integrase mediated cassette exchange.

For targeted insertion at the *Gt(ROSA26)Sor* locus, a PhiC31 integrase mediated cassette exchange approach was adopted using IDG26.10-3 ES cells, which are a (C57BL/6J × 129S6/SvEvTac) F1 ES cell line harboring a PGK promoter driven hygromycin selection cassette flanked by PhiC31 attP sites, positioned within intron 1 of *Gt(ROSA26)Sor*[133]. The targeted insertion of the vector into the *ROSA26* locus ensures reproducible transgene expression. $1 \times 10^6$ IDG26.10-3 ES cells[133] were co-electroporated with 5 μg of either *pCB92-C*05* or *pEx-CAG-KIR2DL1* and 5 μg of pPhiC31o (Addgene) using the Neon transfection system (Thermo Fisher) (3 × 1400 V, 10 ms) and plated on G418 resistant fibroblast feeder layers. After approximately 7 days of selection in 350 μg/ml G418, 16 resistant colonies were isolated per construct, expanded and screened for the correct cassette exchange event at the 5′ and 3′ ends using specific screening primers provided in the Supplementary Data 10 (5′ screen: yields a 280 bp product; 3′ screen: yields a 518 bp product). Correctly integrated ES cell clones were injected into mouse C57BL/6J blastocysts, and the resulting chimeric males were mated to C57BL/6J females and the progeny were screened for germline transgene transmission, normal breeding and appropriate expression of the transgene. The mice were further backcrossed onto C57BL/6J for 4–5 generations. Mice at the same backcrossing generation were used as transgene-negative littermates or controls within each experiment. The NK1.1 antigen is not expressed on the 129 background. The NK1.1 antigen status was always assessed when performing matings and the NK1.1 expression was kept consistent within experiments with mice largely being maintained on a NK1.1-ve background. KIR2DL1-floxed transgenic mice were mated to NCR1-iCre mice (a gift from Veronika Sexl)[46] to obtain KIR2DL1-NCR1-iCre transgenic mice, which had specific expression of KIR2DL1 in NCR1-expressing cells, and referred to as KIR2DL1-expressing mice hereafter. KIR2DL1-NCR1-iCre transgenic mice were crossed to HLA-C*05

transgenic mice to generate double transgenic mice having expression of both HLA-C*05 and KIR2DL1.

For adoptive transfer experiments, HLA-C*05 mice were mated to H2-K$^b$H2-D$^b$ double-knockout mice (a gift of Petter Höglund) to generate HLA-C*05 mice, which lacked expression of H2-K$^b$ and H2-D$^b$. Genotyping of all alleles was done using PCR as described below.

## Cell lines

The B-lymphoblastoid cell line, 721.221 was purchased from the International Histocompatibility Working Group and grown in RPMI-1640 (Thermo Fisher) supplemented with 15% heat-inactivated fetal calf serum (FCS) (Sigma-Aldrich), sodium pyruvate (Thermo Fisher), ʟ-glutamine (Sigma-Aldrich) and primocin (Invivogen). YT cells transfected with *KIR2DL1* (a gift of Dr. Chiwen Chang) were grown in RPMI-1640 supplemented with 10% heat-inactivated FCS, ʟ-glutamine and penicillin−streptomycin (Sigma-Aldrich). HEK 293T cells (ATCC) were cultured in DMEM supplemented with 10% heat-inactivated FCS, L-glutamine and penicillin−streptomycin.

## Genotyping of mice

Genomic DNA from mice and embryos was extracted using E-Z 96 Tissue DNA Kit (Omega Bio-tek) as per the manufacturer's instructions. The expression level of genes of interest for each sample was normalized against mouse housekeeping gene, Glyceraldehyde 3-phosphate Dehydrogenase (*Gapdh*). Genotyping of *HLA-C*05*, *KIR2DL1*, *Ncr1*, and *NK1.1* was performed using real-time quantitative PCR (q-PCR) on a LightCycler 480 II (Roche) under the following conditions: 50 °C for 2 min, 95 °C for 10 min, 40× Quantification mode (95 °C for 15 s, 60 °C for 1 min), Melting Curve mode (95 °C for 15 s, 60 °C for 15 s, 95 °C Continuous Acquisition of 5 per °C), 40 °C for 10 min. qPCR samples were prepared with Power SYBR Green Master Mix (Thermo Fisher) and gene-specific primers as listed in Supplementary Data 10. Comparative CT method ($2^{-\Delta\Delta CT}$) was used for the quantitative analysis of relative gene expression, where ΔCT is the difference between the threshold cycle of gene of interest and *Gapdh*, and ΔΔCT is the difference between the ΔCt of each sample and the positive control sample. Genotyping of H2-K$^b$ H2-D$^b$ double-knockout mice was performed using Touchdown (TD) PCR on a SimpliAmp Thermal Cycler (Thermo Fisher) under the following conditions: 95 °C for 10 min, 17 cycles of (95 °C for 30 s, 60 °C decreased by 0.5 °C every cycle for 40 s, 72 °C for 30 s), 16 cycles of (95 °C for 30 s, 60 °C for 40 s, 72 °C for 30 s), 72 °C for 10 min. TD-PCR samples were prepared with AmpliTaq Gold DNA Polymerase, Gold Buffer, MgCl2, GeneAmp dNTP Blend (Thermo Fisher) and gene-specific primers as listed in Supplementary Data 10. TD-PCR products were electrophoresed in 1.2% Tris-Borate-EDTA (TBE) agarose gels, stained with Midori Green Advance DNA Stain (Geneflow) and imaged with Gel Doc XR + Gel Documentation System with Image Lab software (Bio-Rad).

## Preparation of single-cell suspensions from mouse organs for phenotypic and functional characterization experiments

All organs were placed in R10 medium (details below) on ice until further processing. Single-cell suspensions were counted using an improved Neubauer chamber or Countess I (Thermo Fisher) after dilution in 0.4% trypan blue (Thermo Fisher). All centrifugation steps were done at $480 \times g$ for 5 min at 4 °C unless otherwise stated. R10 constituted of RPMI-1640 media supplemented with L-glutamine, sodium bicarbonate (Sigma-Aldrich), 10% heat-inactivated FCS, 1x Penicillin−Streptomycin and 50 μM 2-Mercaptoethanol (Gibco), freshly added. Spleens were carefully dissociated using the plunger of a 5 ml syringe and filtered through pre-wetted 70 μm cell strainers (Corning). Cells were washed once in R10 and red blood cells lysed in Red Blood Cell Lysing Buffer Hybri-Max (Sigma-Aldrich) for 7 min at room temperature. Cells were washed in R10 and counted. Splenocytes were either used for cell surface and intranuclear flow cytometric

staining or processed for in vitro stimulations. In some experiments, cells were also re-suspended in RLT buffer (Qiagen) supplemented with 2-Mercaptoethanol and stored at −80 °C until RNA isolation. For isolation of cells from lymph nodes (LN), left and right inguinal and axillary LN were carefully dissociated using the plunger of a 5 ml syringe and filtered through pre-wetted 70 µm cell strainers. Cells were washed in R10, counted and then stained as per the flow cytometry protocol. Bone marrow (BM) isolations were done by collecting left and right tibia and flushing bones with 5 ml Hanks Balanced Salt solution (HBSS) (Sigma-Aldrich) using a syringe with a 26G needle. Cells were dissociated by passing the BM through the needle 5–10 times and filtered through pre-wetted 70 µm cell strainers. Cells were washed once in R10 and red blood cells lysed in Red Blood Cell Lysing Buffer Hybri-Max for 7 min at RT. Cells were washed in R10, counted and then stained as per the flow cytometry staining protocol. Isolation of thymocytes were performed by dissociating thymi using the plunger of a 5 ml syringe and filtered through pre-wetted 70 µm cell strainers. Cells were washed in PBS, counted and then stained as per the flow cytometry staining protocol.

## Flow cytometric staining

Surface staining was done in the presence of anti-CD16/32 antibodies to block FcγRII/III receptors using Mouse Fc block (BD Biosciences). Cells were incubated with the Mouse Fc block for 10 min followed by incubation with fluorochrome-conjugated pre-titrated antibodies for 20 min at 4 °C. Cells were washed in FACS buffer (PBS with 2% Heat-inactivated FCS) and fixed in 1x BD cellFIX (BD Biosciences). Samples were acquired on the LSRFortessa (BD Biosciences), Fortessa X-20 (BD Biosciences) using BD FACSDiva software, Cyan ADP (Dako) using summit software or the Attune acoustic focusing cytometer (Applied Biosystems) using Attune NxT software, and analyzed using the FlowJo software (FlowJo). In order to ensure comparability, FluoroSpheres (Dako) and Sphero Rainbow Calibration beads (BD Biosciences) both with defined MEF (molecules of equivalent fluorochromes) were run for each experiment in addition to the samples and used to calculate normalized MFI values. For intracellular staining, cells were fixed, permeabilised and stained using the BD Cytofix/Cytoperm kit (BD biosciences) as per manufacturer's instructions. For nuclear staining of transcription factors, cells were stained using the Foxp3 staining buffer set (eBioscience) according to the manufacturer's instructions. Samples were acquired and analyzed as described above. Details of all antibodies used can be found in Supplementary Data 11.

## Stimulation of splenocytes

For in vitro stimulation of total splenocytes, $2 \times 10^6$ splenocytes were stimulated with 4 µg/ml LPS (Sigma-Aldrich) or 50 U/ml IFNγ (Peprotech) for 24 h in 24-well plates in R10 medium. After incubation, cells were detached by careful pipetting, washed in FACS buffer and stained as per the flow cytometric staining protocol. For in vitro stimulation of NK cells, $1 \times 10^6$ splenocytes were isolated from mice and cultured on Immulon U-bottom plates (Thermo Fisher) coated with antibodies for Ly49D (0.5 mg/ml) (Biolegend) or NKp46 (1 mg/ml) (Thermo Fisher) with or without the presence of anti-KIR antibody (1 mg/ml) (Biolegend), in the presence of 2000 U/ml rhIL-2 (Peprotech) and Brefeldin A (10 µg/ml) (Sigma-Aldrich) for 5 h. Cells were then harvested and stained as per the flow cytometric staining protocol and IFNγ production by NK cells was assessed. Negative and positive controls included cells cultured without any stimulus and those cultured with PMA (250 ng/ml) (Sigma-Aldrich) and Calcium Ionomycin (2.5 µg/ml) (Sigma-Aldrich).

## Adoptive transfer experiments

Adoptive transfer experiments to assess in vivo rejection were performed[134]. Splenocytes were isolated from mice lacking expression of murine MHC class I, i.e., H2K$^{b-}$H2D$^{b-}$ (referred to as KO cells), WT

mice (which were H2K$^{b+}$H2D$^{b+}$) or HLA-C*05-expressing mice, which had been bred to mice lacking expression of murine MHC class I molecules (i.e., HLA-C*05$^+$ H2K$^b$H2D$^{b-}$, referred to as HLA-C*05$^+$ KO cells). Cells were labeled with differential concentrations of CFSE (Sigma-Aldrich) to allow for cellular discrimination as follows: KO cells (0.5 µM of CFSE), WT cells (5 µM CFSE) and HLA-C*05$^+$ KO (5 µM CFSE). Cells were mixed and ~$5 \times 10^6$ cells per population were injected intravenously into either WT or KIR2DL1-expressing recipient mice. Splenocytes were isolated from recipient mice after 20 h and cells were analyzed by flow cytometry using antibodies for HLA-C (B1.23.2) and H2K$^b$ (AF6-88.5.5.3). A small sample of the injection mix was kept and analyzed by flow cytometry for reference. Survival of both HLA-C*05$^+$ KO and KO cells was calculated relative to the WT cells in each experiment using the injection mix as reference. Relative survival of HLA-C*05$^+$ KO compared to survival of KO cells was plotted.

## Isolation of thymic epithelial cells (TECs)

TECs were isolated as per established protocols[135]. In brief, thymus lobes were collected from 6 to 7-week-old mice, were cleaned from fat and connective tissue and the capsule was removed under a dissection microscope. Cleaned tissue was digested in PBS, 0.4 Wunsch Units/ml Liberase (Sigma-Aldrich) and 300 µg/ml DNaseI (VWR) pre-warmed to 37 °C. Tissue was carefully triturated in a step-wise manner. First, after 5 min incubation in 37 °C water bath, the thymus lobes were carefully pipetted up and down 10–15 times using a cut P1000 pipette tip. Tissue fragments were left to settle down for 5 min, the cell suspension transferred into a collection tube containing R10 and cells pelleted. Cells were re-suspended in FACS buffer containing 50 µg/ml DNaseI to prevent cell clumping. Trituration of the remaining tissue fragments was repeated a total of four rounds; with pipette tip openings gradually getting smaller and the last round using an uncut P10 tip. Cells were pooled and counted. An aliquot was taken for flow cytometric staining as a pre-TEC isolation sample. CD45$^+$ cells were depleted using CD45 MicroBeads (Miltenyi Biotech) according to manufacturer's instructions. Cells were filtered sequentially through 70 µm and 40 µm (Corning) cell strainers with 1.8 mg/ml DNaseI added in order to prevent blocking of the MACS columns. The CD45$^-$ cell fraction eluted from the LS columns (Miltenyi Biotech) represented TECs. Cells were counted and flow cytometric surface marker staining was performed as described above.

## Lentiviral transduction and in vitro co-culture assays

The *HLA-C*05* DNA construct (used to make transgenic mice) and a similarly designed *HLA-C*07* control DNA construct were sub-cloned into the lentiviral pHRsinUbEm expression plasmid[136] (a gift from J.M. Boname/ P.J. Lehner, University of Cambridge). To confirm that the presence of the H2-K$^b$ α3 domain in the HLA-C construct did not hamper recognition of the HLA-C molecule by KIR2DL1, two additional constructs, C*05 (α3-H) and C*07 (α3-H) constructs, were made. In the C*05 (α3-H) and C*07 (α3-H) DNA constructs, the α3 domain, transmembrane and cytoplasmic regions, and hence the exons 4–8 of the HLA-C molecule were taken from the respective *HLA-C* genes. The lentiviral HLA-C expression plasmids were co-transfected with the vesicular stomatitis virus-G envelope plasmid pMD2.G (Addgene) and packaging plasmid psPAX2 (Addgene), containing HIV-1 Gag and Rev, into HEK 293T cells to package lentiviral particles. Viral titers were determined by serial dilution and transduction of HEK 293T cells. 721.221 cells were transduced with the packaged lentivirus at a multiplicity of infection of 20, in the presence of polybrene (Santa Cruz Biotechnology), added at a final concentration of 8 µg/ml. Cells were harvested 72 h post transduction and HLA-C expression determined by staining with anti-HLA (W6/32) (Biolegend) antibody. The HLA-C transduced 721.221 cells were then co-cultured with YT-KIR2DL1 cells at a ratio of 0.5:1T:E for 6 h and IFNγ production by the YT-KIR2DL1 cells was assessed by staining with antibodies for IFNγ (XMG1.2) and

KIR2DL1 (HP-MA4). In addition, controls where the empty-vector transduced 721.221 cells or non-transduced 721.221 cells were co-cultured with YT-KIR2DL1 cells, or where YT-KIR2DL1 cells were cultured with PMA and Calcium Ionomycin were included.

## Isolation and culture of trophoblast cells

Trophoblast cells were isolated from implantation sites from age-matched pregnant female mice at gd12.5[137]. All placentas from the litter from a mouse were pooled and minced in RPMI. For every 6–8 placentas, 25 ml digestion buffer containing HEPES (20 mM) (Gibco), sodium bicarbonate (0.35 g/L) (Gibco), collagenase type I (1 mg/ml) (Sigma-Aldrich), DNAse I (20 μg/ml) in RPMI supplemented with Glutamax was used and incubated at 37 °C for 1 h. Cells were centrifuged at $500 \times g$ for 5 min, and the pellet was re-suspended in 4 ml of 25% Percoll (Sigma-Aldrich) and layered on top of 4 ml of 40% Percoll, followed by centrifugation at $850 \times g$ for 20 min at 4 °C without brake. Trophoblast cells were removed from the interface, washed with PBS, incubated with ACK lysis buffer followed by another wash in PBS. Cells were plated at $1 \times 10^5$ cells/well in 96-well U-bottom plates and cultured in RPMI supplemented with Glutamax, 10% heat-inactivated FCS, sodium pyruvate (Thermo Fisher), penicillin/streptomycin, L-glutamine and 50 μM 2-mercaptoethanol for 4 days, with media replenished after 2 days. On day 3, half the cells were stimulated with LPS (10 μg/ml) (Sigma-Aldrich) and IFN-γ (500U/ml) (Peprotech), and the other half were cultured without stimulus. Cells were then harvested using Accutase cell dissociation reagent (Fisher Scientific) after 24 h and stained as per the flow cytometric staining protocol.

## Enzyme-linked immunosorbent assay (ELISA)

Mouse VEGF R1/Flt-1 (R&D systems), PlGF-2 (R&D systems) and Endoglin/CD105 (R&D systems) were measured in plasma samples collected from pregnant mice on gd18.5. For collection of plasma, murine blood was collected in EDTA-coated tubes followed by centrifugation at $2000 \times g$ for 15 min at room temperature. Supernatants (plasma) were collected, aliquoted and stored at −80 °C. Urine samples were also collected from pregnant female mice at gd18.5 (just prior to culling the mice for measurement of fetal weights), and aliquoted and stored at −80 °C. Albumin and creatinine concentrations in the urine were determined by an Albumin ELISA (Bethyl Laboratories) and a Creatinine colorimetric assay (Cayman Chemical), respectively. The ratio of urinary albumin to creatinine was calculated and considered as a measure of proteinuria.

## RNA isolation, cDNA preparation, and Q-PCR

Total RNA was isolated from cells using the RNeasy isolation kit (Qiagen), and cDNA prepared using the Quantitech Reverse Transcription kit (Qiagen). SYBR-green-based quantitative PCR assays were deigned and optimized for *HLA-C* and *Hprt*. Samples collected from tissue such as dissected implantation sites from pregnant mice or organs were cut into small pieces and immersed in RNAlater (Qiagen) as per manufacturer's instructions, prior to processing for RNA isolation. The sequences of the primers used for the assays are provided in Supplementary Data 10. Standard curves for each of the assays were performed using serial dilutions of cDNA and amplification efficiencies were determined. Relative expression was expressed as $2^{-\Delta Ct}$, where ΔCt is the difference of the cycle threshold between the transcript of the gene of interest and the reference gene transcript.

## RNA in situ hybridization (RNA-ISH)

RNA-ISH was performed on mouse implantation site tissues isolated from pregnant mice on gd9.5. Implantation sites were embedded in OCT (Thermo Fisher) and snap frozen in isopentane (VWR) and dry ice. Serial 10 μm coronal sections were cut on a cryostat, kept at −20 °C to dry for an hour and then stored at −80 °C. Sections were hybridized with mRNA probes specific for the *Ccl1*, *Gzme/d/g* and *Anxa2* genes

(ACD). Following probe hybridization and amplification, mRNA was detected using the RNAscope 2.5 HD Duplex Assay (ACD), as per the manufacturer's instructions. For quantification of stainings, tissue from four independent implantation sites were used per mouse group with multiple sections from the same region of the tissue used from each implantation site. Image analysis was completed by the Pharma Services group at Indica Labs (Albuquerque, NM, USA). Slides were scanned at 40x magnification on the Aperio Scanscope Turbo AT2 Slide Scanner (Leica Biosystems) and digital images imported into HALO Image Analysis Platform (Indica Labs, Albuquerque, NM USA). Collaboration between the two groups occurred through the use of a cloud-based HALO Link Image Management system. Image analysis was based on teal-red duplex RNAScope images counterstained with hematoxylin (Sigma-Aldrich) for nuclei detection. HALO Link was utilized to import sample metadata and for researcher annotation of samples for regions of interest (ROIs). Annotations were used to identify individual tissue sections on separate layers of analysis. Multiple HALO algorithms were utilized for analysis: ISH v3.3.9 (RNA probe identification) and Multiplex IHC v2.1.1 (cytoplasmic and nuclear staining localization). The ISH algorithm was optimized to detect positive intensity of individual "spot-like" probes of *Anxa2* and *Ccl1*. *Anxa2* displayed the expected "spot-like" staining pattern of RNA-Scope while *Ccl1* displayed a stronger "spot-like" staining pattern than whole cell or nuclear staining, and both of these were accurately identified using ISH. In addition to the ISH algorithm we also utilized our Multiplex IHC algorithm because the staining pattern for *Gzme/d/g* was not a typical probe-like RNA stain. Therefore, the Multiplex IHC algorithm more accurately quantified this probe. The Multiplex IHC algorithm was optimized to detect positivity within the nuclear compartment of *Gzme/d/g*. The ISH module calculated the Probe Copies/cell for *Anxa2* and *Ccl1*, and the Multiplex IHC module calculated the overall % positive cells for *Gzme/d/g*.

## Blood pressure measurement in mice

Systolic blood pressure was measured in mice using a non-invasive blood pressure analyzer, BP-2000 Blood Pressure Analysis System (Bioseb), that uses tail-cuff transmission photoplethysmography. The restraint platform was kept at a temperature of 37 °C. Baseline blood pressure measurements for each mouse were taken prior to using the mice for timed-matings. Baseline measurements required recording of valid measurements for at least 3 days for each mouse with a minimum of 4 valid readings on each day of measurement. During the course of gestation, measurements were taken every alternate day from gd2.5 until gd14.5 and then every day from gd15.5 until gd18.5. Post-partum measurements were taken daily from day 1 post-partum until 6–7 days post-partum. Inclusion of measurements during pregnancy and post-partum for each mouse required recording of a minimum of 4 valid readings on each day of measurement. Collected data were analyzed with the BP-2000 Analysis Software (Bioseb) and the change in blood pressure for each mouse was normalized to its baseline.

## Perfusion of uteroplacental vasculature for Micro-CT imaging and vascular segmentation analysis

The silicon rubber injection compound, Microfil (Flow Tech) was injected into the uteroplacental vasculature of mice for micro-CT imaging using established protocols[138]. In brief, pregnant mice at gd10.5 were given terminal anesthesia using pentobarbitone and intracardiac perfusion was performed using a Heparin solution (100 IU/ml) in PBS. A catheter was placed in the descending thoracic aorta and used to clear the lower body vasculature of blood and perfused with heparinised PBS. The contrast agent, Microfil was then gently infused by hand on a slow flow rate until the blue color of the contrast agent was visible through the capillary bed and sufficiently vented through the right atrium. The inferior vena cava was then immediately tied off and the vasculature was kept pressurized using a

syringe that contained the contrast agent to sustain vessel inflation, until the Microfil polymerized. The uterus was then removed, immersed in formalin (Sigma-Aldrich) overnight and the implantation sites were mounted in 1% agarose for micro-CT imaging. Micro-CT imaging was performed on Skyscan 1172 (Bruker) at 40 kV, 250 µA, 750 ms exposure time and a pixel size of 4.73 µm, and the three-dimensional (3D) images were reconstructed using Nrecon (Micro Photonics). As the uteroplacental vasculature was difficult to segment automatically due to the unique nature of the tightly coiled spiral arteries, the micro-CT datasets were manually segmented slice by slice using the seg3D software[139]. Each acquired image stack was first downscaled by 50% (to typical dimensions $900 \times 900 \times 1000$) and 3D median filtering was applied (radius 2 voxels, isotropic). Each voxel of the resulting stack was classified as background, vessel or transitional using multi-Otsu thresholding. With the aid of these pre-assigned non-overlapping classes, assisted manual segmentation was performed using seg3D, segmenting the volume into spiral arteries, radial arteries, maternal canals, uterine arteries, and background. The first step of the automated analysis was to translate the voxel-based information in the segmented image stacks into a graph structure that is better suited for quantification algorithms. The uterine, spiral, radial and maternal canal vessels were combined into a single class and skeletonisation was performed using homotopic thinning[140]. The skeletonized network was then organized into segments, where a segment denotes the ordered chain of vertices that run between two adjacent junctions, or a junction and a terminal. Spurious segments that may result from noisy segmentation were detected using a custom routine and removed from the network[141]. All subnetworks (a collection of segments that are connected to each other) were detected and the smallest subnetworks below the threshold (<30 segments) were removed. Vessel class labeling from the segmentation was automatically transferred to the segments. As the skeletonisation process places nodes at discrete voxel centers, the total number of nodes in a given network depended on the total length of the vessels in it. To allow inter-sample comparisons, we chose to normalize the number of total nodes in each network to 10,000 by resampling along each segment. The distance between each adjacent pair of nodes was kept approximately constant, subject to the constraint of a discrete number of voxels along a segment. At each node, the local radius of the vessel segment was estimated using Rayburst algorithm[142]. Furthermore, to refine the voxel-based quantification of radius, a sequence of synthetic vessel-like image stacks with known radii were created and nonlinear regression was used to calibrate the Rayburst estimates into sub-voxel quantities. Using the Rayburst kernel, the location of each skeleton node was also adjusted towards the true vessel centreline on a sub-voxel basis, to reduce the staircase artefact. Vessel morphological quantifications were performed either on a nodal basis for radii calculations, or segmental basis for calculation of length and volume, for vessels >10 µm in radius and length. For segmental quantification, the summed segment length and volume were calculated using a piecewise linear approximation. Flow conductance of each segment was calculated as the reciprocal of Poiseuille resistance, under the assumption that the flow will remain laminar. Given the typical inlet vessel dimensions, this assumption was regarded as appropriate. However, the calculation of total spiral arteries conductance requires further assumptions about boundary conditions. For this, the terminal vessels of the spiral sub-network were detected and their proximity to the radial or canal vessels were evaluated, thereby classifying every terminal as being radial- or canal-adjacent. All radial-connected spiral arteries were assumed as flow inlets and all canal-connected arteries, as outlets. In general, the number of inlets and outlets are not consistent between different samples. We calculated the total spiral network resistance by imposing a pressure gradient between the inlets and outlets of 1 kPa and calculating the total outflow using a network-Poiseuille formulation, as $R = \triangle P / Q$. Owing to the linearity of the flow model, the resistance remains constant regardless of the actual pressure gradient applied. For analysis, flow conductance is quoted, which are reciprocal of the resistance (the greater the conductance, the easier the fluid to flow).

## Isolation of cells from implantation sites and FACS sorting for scRNA-seq studies

Isolation of cells from the implantation sites were done with enzymatic digestion using Liberase[143]. In brief, uterine horns were collected in cold HBSS containing $Ca^{2+}$ and $Mg^{2+}$ (Thermo Fisher) from pregnant mice and cleared of mesometrial fat. The tissue was collected from the entire mesometrial side of the implantation site leaving behind the developing fetus[144,145]. All implantation sites from a litter were pooled and processed together. The tissue was finely minced, incubated in HBSS solution containing 0.28 Wunsch Units Liberase and 30 µg/ml DNase for 30 minutes at 37 °C with agitation, washed with $Ca^{2+}/Mg^{2+}$-free PBS containing 5 mM EDTA (Thermo Fisher) and centrifuged. Cells were incubated in $Ca^{2+}/Mg^{2+}$-free PBS containing 5 mM EDTA for 15 min at 37 °C, followed by filtering through a 70 µm cell strainer, red blood cell lysis and washing in PBS. Cells were then filtered again through a 70 µm cell strainer and counted.

For isolation of total unsorted cells from the implantation site, half of the isolated cells were taken and depleted of dead cells by labeling with the Dead cell removal microbeads (Miltenyi Biotech) as per the manufacturer's instructions. The viable cell fraction eluted from the MS columns (Miltenyi Biotech) was counted and processed for droplet-based scRNA-seq (below).

For isolation of uNK cells, half of the isolated cells were surface stained using the flow cytometric staining protocol for antibodies against CD45 (30-F11), CD3 (145-2C11), CD19 (eBio1D3), TCRb (H57-597), NKp46 (29A1.4), CD49a (Ha31/8), CD49b (DX5), KIR2DL1 (HP-MA4), CD122 (TM-b1) along with either 7-AAD (eBioscience) or a Live/Dead viability dye (Biolegend). For SMART-Seq2 processing, single cells were FACS sorted into 96-U-well plates into either trNK (NCR1$^+$CD3ε$^-$TCRb$^-$CD19$^-$CD45$^+$CD122$^+$CD49a$^+$) or cNK (NCR1$^+$CD3ε$^-$TCRb$^-$CD19$^-$CD45$^+$CD122$^+$CD49b$^+$) cells. Single cells were directly sorted into wells containing 5 µl of pre-chilled TCL buffer (Qiagen) containing 1% 2-mercaptoethanol[57]. Each plate consisted of a negative control with no cells and a positive control with 15 cells sorted in one well. 96-well plates containing the sorted cells were then immediately centrifuged and frozen directly on dry ice prior to storage at −80 °C. For droplet-based scRNA-seq, total uNK cells (NCR1$^+$CD3ε$^-$TCRb$^-$CD19$^-$CD45$^+$CD122$^+$) were sorted into Eppendorf tubes containing PBS with 0.04% BSA and counted and processed as described below. Samples were sorted on the FACS Aria II or FACS Aria III (BD Biosciences) using BD FACSDiva software.

## Full-length scRNA-Seq

SMART-Seq2 scRNA-seq libraries were prepared from trNK and cNK cells sorted in 96-U-well plates[56,57] with some modifications in the protocol[146]. Briefly, RNA was purified from single-cell lysates using 2.2x RNAClean XP beads (Beckman Coulter), eluted in 4 µl of master mix containing 1 µl of 10 mM dNTPs (Thermo Fisher), 0.1 µl of 100 µM 3′ oligodT RT primer, 0.1 µl of 40 U/µl RNase Inhibitor (Clontech Takara) and 2.8 µl of 1 M Trehalose (Life Sciences Advanced Technologies), followed by incubation at 72 °C for 3 min and placing on ice. The remaining reverse transcription mix, containing 2 µl of Maxima RT buffer (Thermo Fisher), 0.1 µl of 100 µM TSO oligo, 0.25 µl of 40 U/µl RNase Inhibitor, 0.1 µl of 200 U/µl Maxima H Minus Reverse Transcriptase (Thermo Fisher), 3.45 µl of 1 M Trehalose and 0.1 µl of 1 M MgCl$_2$ (Sigma-Aldrich) was added to each well and incubated for 50 °C for 90 min followed by 85 °C for 5 min. The cDNA was then amplified using 12.5 µl of KAPA HiFi HotStart ReadyMix (KAPA Biosystems) and 0.05 µl of 100 µM ISPCR primer and incubated at 98 °C for 3 min

followed by 23 cycles of 98 °C for 15 s, 67 °C for 20 s, 72 °C for 6 min, and a final extension at 72 °C for 5 min. PCR products were purified using 0.7x AMPure XP SPRI beads (Beckman Coulter) and eluted in 20 μl of TE buffer (Qiagen). DNA quantification was done using the Qubit dsDNA HS Assay Kit (Thermo Fisher) and DNA fragment size was assessed using the High Sensitivity DNA BioAnalyzer kit (Agilent). cDNA concentrations were normalized and 0.125 ng of each cDNA was used in a quarter volume of a Nextera XT DNA library preparation kit (Illumina), following pooling of equal volumes from each well and purification with 0.6x AMPure XP SPRI beads. Final libraries were again assessed for DNA concentration using the Qubit assay and fragment size using the Bioanalyzer. All SMART-Seq2 Libraries were sequenced using a NextSeq 500/550 High Output v2 kit 75 cycles (Illumina) on a Nextseq 500 (Illumina).

## Droplet-based scRNA-Seq
For droplet-based scRNA-seq, 7000 cells were loaded onto the Chromium Single-Cell 3′ Chip A (10x Genomics) and processed through the chromium controller to generate Gel beads in Emulsion. RNA-seq libraries were prepared using the Chromium Single-Cell 3′ Library & Gel Bead Kit v2 (10x Genomics) as per the manufacturer's instructions. Libraries were sequenced on an Illumina Hi-Seq.

## Statistical analysis
Data were analyzed using Prism 8 (GraphPad Software) or RStudio. Statistical details of experiments including sample size can be found in the figure legends. To statistically estimate the effect of genotype on fetal or placental weight, a linear mixed-effects model was used with transgenic genotype mating combination as a fixed effect and clustering of observations within a litter as a random effect. Similarly, for estimation of the effect of genotype on spiral artery diameter and segment length, a linear mixed-effects model was used with transgenic genotype mating combination as a fixed effect and clustering of observations from a specific implantation site as well as clustering of observations within a litter from a mouse as random effects. As the segment length data followed a log-normal distribution, the mixed-effects model for assessing effect on segment length was run on log-transformed length values. For comparison of two-group datasets with non-Gaussian distribution, the non-parametric Two-tailed Mann–Whitney U-test was used. Values were expressed as mean ± S.E.M. p-values lower than 0.05 were considered statistically significant. For all the statistical analysis: $*p < 0.05$, $**p < 0.01$, $***p < 0.001$, $****p < 0.0001$.

## scRNA-Seq data preprocessing
For full-length scRNA-Seq SMART-Seq2 data, raw sequencing data was demultiplexed to fastq files using bcl2fastq (v.2.17.1.14) and aligned to a custom reference using RSEM (v.1.2.8); the custom reference included the mm10 genome and the sequence for the human genes KIR2DL1 and HLA-C*05 (as described in the transgenic mice generation above), and was generated using rsem-prepare-reference. Quality control was assessed and summaries generated using STAR. We used RSEM (v1.2.8) to quantify gene counts and TPM (we converted TPM to TP100K after initial quality control filtering, described below).

For droplet-based scRNA-Seq data, raw sequencing data was demultiplexed to fastq files with cellranger mkfastq (version 2.1.0, 10x Genomics) and reads were aligned and count matrices were generated with cellranger count (version 2.0.1). A custom reference was generated to include the HLA-C*05 and KIR2DL1 transgenic construct sequences into the mm10 reference using cellranger mkref (version 2.1.0). The alignment of HLA-C*05 sequence was limited to the human DNA sequence region to minimize the risk of misalignment with the mouse H-2K$^b$ gene.

## scRNA-Seq quality control
For full-length scRNA-seq, lower-quality cells were removed if they met any of the following criteria: (1) $\log_{10}(counts) < 5$, (2) number of expressed genes <1000 or >7000, or (3) average housekeeping gene (Supplementary Data 2) expression (TPM) > 1. In addition, cells that met any of the following criteria were also removed (to ensure cells meet FACS expectations): (1) Cells with HLA-C*05 expression <2 in C*05/KIR × WT mating group; (2) Cells with HLA-C*05 expression >2 in WT × C*05 mating group, (3) Cells with KIR2DL1 expression <2 in KIR × C*05 and C*05/KIR × WT mating groups; (4) Cells with KIR2DL1 expression >2 in WT × C*05 mating group; (5) Cells with Cd3e expression >3; (6) In sorted cNK cells, cNK cells with Itga1 expression >1. Genes expressed in three or fewer cells were removed. Final expression values were normalized to TP100K and log transformed.

For droplet-based scRNA-seq, the feature-barcode matrix cellranger count was used, but with force-cells = 6000 to ensure all droplets containing cells are included. Low-quality cells were removed by filtering any cells with either (1) less than 500 genes, (2) less than 1000 UMIs, or (3) more than 10% of UMIs mapped to mitochondrial genes. Genes expressed in three or fewer cells were removed. The final expression values were normalized to a total UMI per cell of $1 \times 10^4$ and log transformed.

We used Scrublet (v.0.2.1), with expected_doublet_rate = 0.06, to predict doublets[147]. In the NK cell sorted droplet-based data we removed one cluster with high doublet probability as well as one cluster of non-NK cells, and then re-analyzed the remaining data. In the unsorted droplet-based data we excluded from visualization and analysis cells from two clusters with high doublet scores, but did not reanalyze the data following exclusion.

## Dimensionality reduction, clustering, and visualization
In both full-length and droplet-based scRNA-seq, human genes were removed from the expression matrix prior to dimensionality reduction. We did not batch-correct by mouse because mice and genotype were evenly distributed across clusters (Supplementary Figs. 4c, f, i, l and 7e), adjusted rand index between sample and cluster = 0.49% for full-length NK data, 0.88% for droplet-based NK data, and 1.21% for the droplet-based unsorted data).

For both the full-length and droplet-based scRNA-seq data, highly variable genes were selected using FindVariableGenes (default parameters) in Seurat[148] [v.2.1]. For the full-length data, the expression of each gene was centered and scaled to have a mean of zero and a standard deviation of one; for droplet-based data, the expression of each gene was centered to have a mean of zero.

In both full-length and droplet-based datasets, initial dimensionality reduction was performed by principal component analysis (PCA) using RunPCA in Seurat (v.2.1)[148], and Louvain graph-based clustering[60] was performed using FindClusters (based on the top 15 PCs, with clustering resolutions of 0.8, 0.4, and 0.6 for the full-length NK, droplet-based NK, and unsorted droplet-based data, respectively; other parameters, including $k = 30$ for $k$-NN, were set to the default). Cell profiles were visualized using a Uniform Manifold Approximation and Projection (UMAP)[149] embedding of the top 15 PCs, with min-dist = 0.5, spread = 1.0, number of neighbors = 15, and the Euclidean distance metric.

The statistical significance of genotype proportions within each cluster was tested using a Dirichlet-multinomial regression[61]. Mouse batch effects were assessed within each cluster by calculating the rand index using the Clues package in R.

## Differentially expressed genes, annotating cell subsets, and building gene signatures
Subset-specific marker genes (all datasets) and cNK/trNK-specific (full-length data) differentially expressed genes were identified by a

Mann–Whitney *U*-test using the FindAllMarkers function in Seurat (v.2.1), with test.use = wilcox, logfc.threshold = log(1.5) and min.pct = 0.2; the Benjamini–Hochberg false-discovery rate (FDR) was calculated using p.adjust in R (method = Benjamini & Hochberg). Cell subsets were manually annotated using cell subset marker genes and expression of relevant published signatures[88,128,150–154]. For the two NK cell scRNA-seq datasets (full-length and droplet-based) and some cell subsets from the unsorted droplet-based data, novel cell subsets are described by their expressed marker genes.

Within each subset, we tested for differentially expressed genes between FGR and CTR1, FGR and CTR2, and CTR1 and CTR2 using: (1) a pseudobulk differential expression analysis (Supplementary Data 5 and 7) and (2) a mixed-effects Poisson regression model (Supplementary Data 12 and 13). In both approaches, genes were only tested for differential expression if they were expressed in at least 10% of cells within one of the tested conditions for a given cell subset. For the pseudobulk approach, we summed counts per mouse within a subset and then performed differential expression analysis using DESeq2 (v.1.26, design formula: gene ~ condition)[155]. We used the fdrtools (v.1.2.16) package[156] in R to calculate FDR based on the empirical null, with the DESeq2 Wald statistic as input. For the mixed-effects Poisson regression model[157], we used the glmer function from the lme4 package (v.1.1-26)[158] in R with family = poisson() and the following design formula: gene ~ condition + offset(log(total_UMI/mean_UMI)) + (1|mouseID). We calculated FDR using p.adjust in R (method = Benjamini & Hochberg). FGR-unique expression changes were defined as genes that were differentially expressed in the FGR vs. CTR1 (FDR < 0.05), but not differentially expressed in CTR1 vs. CTR2 (FDR < 0.05). Volcano plots of differentially expressed genes were made using the R package, EnhancedVolcano (v.1.4.0)[159].

Gene signatures for the cNK and trNK cells and for NK cell subsets were defined as the union of the top 25 differentially expressed genes (sorted by *p*-value) and the top 25 differentially expressed genes sorted by log fold change (Supplementary Data 1 and 3). The proliferation signature was previously reported[154]. Gene signatures were scored in each cell using AddModuleScore in Seurat[148] (v.2.1 for scoring proliferation and v.3.2.2 for scoring NK cell subsets).

### Topic modeling

Topic models were fit to the droplet-based sorted uNK UMI counts using the FitGoM() function from the CountClust (v.1.6.1) R package[62–64]. To select the number of topics, *K*, models are fit for each of a range of *K* (*K* = 4:22 in increments of two) with a tolerance parameter (tol) of tol = 0.1 and tol = 0.01. The Bayesian information criterion (BIC), estimated likelihood, and Akaike information criterion (AIC) were computed using compGoM() for each model. The final value of *K* (*K* = 16) was chosen by considering the BIC curve (seeking a *K* for which the BIC was either minimizing or decreasing slowly) and the biological relevance of the identified gene programs. The final value of tol = 0.01 was chosen primarily by the biological relevance of the identified gene programs. Top scoring genes per topic were defined using the extractTopFeatures function.

Topics significantly changed in FGR were selected by examining four criteria: (1) the sign of the KS-statistic; (2) the Benjamini–Hochberg FDR following the KS-test; (3) the fold change direction and magnitude of the mean; and (4) the fold change direction and magnitude of the median. The ks.test and p.adjust (with method = Benjamini & Hochberg) functions in R were used to calculate the FDR for FGR vs. CTR1 and FGR vs. CTR2 under the KS-test. The KS-statistic was calculated with the ks.test function in R as *cumsum(ifelse(order(w) <= n.x, 1/n.x, -1/n.y))*, where *n.x = length(x)*, *n.y = length(y)*, and *w = c(x,y)*. The final statistic was reported as the maximum or minimum of the above equation, whichever had the larger absolute magnitude. We examined the above four criteria using 100% of cells as well as cells with cell weights in the top 75%, 50%, and 25% for each topic; we report metrics based on cells with weights in the top 75% as this retained the most number of cells without a skew towards cells with very low weights in a topic.

EnrichR[160,161] and ingenuity pathway analysis (IPA) (Qiagen) were used to calculate pathway enrichment. Enrichment of canonical pathways, diseases and functions was performed in IPA using gene lists of FGR-unique expression changes (defined as genes that were differentially expressed in FGR vs. CTR1 but not differentially expressed in CTR1 vs. CTR2 within each subset; above); the background was the intersection of genes that were in IPA with the genes that were expressed in the cell subset in at least 10% of the cells from either condition.

### Cell–cell interactions

Mouse genes were mapped to their human genes orthologs using the Human Phenotype Ontology (http://www.informatics.jax.org/downloads/reports/HMD_HumanPhenotype.rpt). CellPhoneDB[88] was then used to identify individual significant ligand–receptor interactions between subsets in the unsorted droplet-based scRNA-Seq data with cellphonedb method statistical_analysis meta.txt counts.txt --result-precision = 3 --iterations = 1000, where meta.txt is cell subset assignments, counts.txt is the normalized counts data for either the WT × C*05 or KIR × C*05 matings, result-precision = 3 indicated the number of decimal points for the reported *p*-value, and iterations was the number of iterations for the statistical test.

The number of significant interactions was summed for each subset pair, and interactions were visualized using Cytoscape v3.8.2[162]. Only subset pairs with at least 50 interactions were connected on the Cytoscape map, and line thickness corresponded to the number of significant interactions.

### Reporting summary

Further information on research design is available in the Nature Research Reporting Summary linked to this article.

## Data availability

The raw and processed gene expression data in this study have been deposited in the Gene Expression Omnibus database under accession code GSE202983. The processed gene expression data are also available on the Single-Cell Portal (https://singlecell.broadinstitute.org/single_cell/study/SCP1312/). All other data are available in the article and its Supplementary files or from the corresponding author upon reasonable request. Source data are provided with this paper.

## Code availability

The code used for analysis of the single-cell RNA-sequencing data presented in this paper is available on https://github.com/GKaur101/Fetal_Growth_Restriction and on Zenodo at https://doi.org/10.5281/zenodo.6538595. The code used for the vascular segmentation analysis presented in this paper is available on https://github.com/jack-kcl/vascular-processing and on Zenodo at https://doi.org/10.5281/zenodo.6553163.

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

## Acknowledgements

L.F. was supported by the Wellcome Trust (grant no. 100308/Z/12/Z), Danish National Research Foundation, Takeda, the Medical Research Council (grant no. MC_UU_12010/3), the Oak Foundation (grant no. OCAY-15-520) and the NIHR Oxford BRC. G.M. was supported by the Wellcome Trust (grant no. 100956/Z/13/Z) and the Li Ka Shing Foundation. A.R. was an Investigator of the Howard Hughes Medical Institute. This project was also funded in part by the Klarman Cell Observatory. O.A. was supported by the NIDDK (5RC2DK116691). This project has been funded in part with federal funds from the Frederick National Laboratory for Cancer Research, under Contract No. HHSN261200800001E. The content of this publication does not necessarily reflect the views or policies of the Department of Health and Human Services, nor does mention of trade names, commercial products, or organizations imply endorsement by the U.S. Government. This Research was supported in part by the Intramural Research Program of the NIH, Frederick National Lab, Center for Cancer Research. We would like to thank V. Sexl (University of Veterinary Medicine of Vienna) for the NCR1-iCre transgenic mice, P. Höglund (Karolinska Institutet) for H2-Kb H2-Db knockout mice, B. Davies (Wellcome Trust Centre for Human Genetics, University of Oxford) for transgenic mouse generation services, D. Bowman, A. Ortiz, and S. Jerman (Indica Labs, Albuquerque, NM, USA) for RNAscope image analysis services, L. Gaffney for figure edits, G. Holländer and M. Deadman (Dept. of Pediatrics, University of Oxford) for help with TEC isolations, C. Smillie, S. Simmons, A. Haber (Broad Institute of MIT and Harvard), and A.-C.Villani (Harvard Medical School) for brainstorming

and discussions, G. Douglas and V. Rashbrook (Radcliffe Dept. of Medicine, University of Oxford) for help with blood pressure measurement in mice, A. Vernet (University of Oxford) for helping with micro-CT scanning, R. Kuehn (Helmholtz Center Munich) for help with transgenic vector design. We also thank the MRC WIMM core Transgenic service team for providing cryopreservation services. We would like to acknowledge S.-A. Clark, C. Waugh, K. Clark, and P. Sopp in the flow cytometry facility at the MRC WIMM for providing cell sorting services. The flow cytometry facility is supported by the MRC HIU; MRC MHU (MC_UU_12009); NIHR Oxford BRC; Kay Kendall Leukaemia Fund (KKL1057), John Fell Fund (131/030 and 101/517), the EPA fund (CF182 and CF170) and by the MRC WIMM Strategic Alliance awards G0902418 and MC_UU_12025.

## Author contributions

Conceptualization, G.K. and L.F.; Methodology, G.K., C.B.M.P., O.A., S.J.R., M.H., A.S., I.A.-D., and G.M.; Software, C.B.M.P., G.K., J.L., O.A., S.J.R., M.H., and A.S.; Formal Analysis, C.B.M.P., G.K., M.A., J.L., O.A., S.J.R., M.H., A.S., K.E.A., C.A.E.D., and J.L.D.; Investigation, G.K., C.B.M.P., M.A., S.B.K., K.E.A., C.A.E.D., H.G.E., L.T.N., D.A.D., A.E.N., L.T.J., T.R.B., and E.S.; Writing—original draft, G.K. and C.B.M.P.; Writing—review and editing, G.K., C.B.M.P., O.A., A.R., and L.F.; Funding acquisition, L.F., A.R., O.R.-R., and G.K.; Supervision, L.F., A.R., O.R.-R., M.C.; Project administration, G.K., C.B.M.P., O.R.-R., and L.F.

## Competing interests

A.R. is a co-founder and equity holder of Celsius Therapeutics, an equity holder in Immunitas, and was an SAB member of ThermoFisher Scientific, Syros Pharmaceuticals, Neogene Therapeutics and Asimov until July 31, 2020. From August 1, 2020, A.R. is an employee of Genentech and has equity in Roche. O.R.-R. is an employee of Genentech as of October 19, 2020 and has equity in Roche. O.A., O.R.-R. and A.R. are co-inventors on patent applications filed by the Broad Institute for inventions related to single-cell genomics, such as in PCT/US2018/060860 and US provisional application no. 62/745,259. G.M. is a director of and shareholder in Genomics plc and a partner in Peptide Groove LLP. The remaining authors declare no competing interests.
