## [Peer Review File · Nature Communications]

Fetal growth restriction through parental and fetal immune gene variation and intercellular communications cascadeREVIEWER COMMENTS

Reviewer #1 (Remarks to the Author):

In this manuscript by Kaur et al., the authors have set up an elegant system to study the contribution of interacting human immune genes HLA-C and KIR to fetal growth restriction (FGR). Genetic association studies in humans have implicated combinations of maternal KIR A haplotype and paternal group 2 HLA-C alleles with increased risk of pregnancy complications such as FGR. The interpretation is that inhibition of maternal natural killer (NK) cells by KIR2DL1, which binds to group 2 HLA-C alleles (C2) on fetal cells, restrains their pro-angiogenic function at the implantation site. However, the extensive polymorphism and complexity of both KIR and HLA genes make it difficult to tease apart the role of specific interacting gene pairs in FGR in humans.

The authors exploited a humanised transgenic mouse model system to examine a single pair of HLA-C and KIR risk alleles. These are the KIR A haplotype KIR2DL1 and the group 2 HLA-C*0501. Remarkably, due to this interaction between an inhibitory KIR and its cognate HLA-C, they uncovered evidence of decreased vascular remodeling at the maternal fetal interface in mice during gestation. In addition to a ~12% decrease in weight, there were changes in spiral artery diameter and length that resulted in reduced vascular conductance and increased resistance to blood flow. It is noteworthy that FGR was observed only when there is expression of maternal KIR2DL1 and of the HLA-C*05 inherited paternally by the fetus.

They further carried out transcriptional profiling of this site in the mouse uterus to show that engagement of this receptor-ligand pair has consequences that impact gene programs relating to placentation not only in uterine NK (uNK) cells but also in other cell types at this site. Their single cell atlas at midgestation (gd9.5) identified 7 uNK subsets that showed features of increased lymphocyte activation and reduced ECM tissue remodeling programs. These FGR associated gene changes were not limited to uNK but were also seen in other cell types such as T and B lymphocytes, macrophages, stromal cells and trophoblasts. Despite significant differences in murine and human reproduction, such as the anatomy of implantation, this study highlighted surprising similarities between the human and murine maternal-fetal interface in terms of cellular crosstalk. They used the CellPhoneDB App to identify receptor-ligand pairs and showed a net increase in intercellular communication when KIR2DL1-HLA-C*0501 interactions occur.

This is a carefully controlled study and the data are presented clearly. These findings have important implications for the study of HLA-KIR disease associations in general as they provide an in vivo correlate of human risk gene combinations in pregnancy that for obvious ethical reasons are impossible to understand at a mechanistic level in humans.

Reviewer #2 (Remarks to the Author):

Comments Manuscript Number:
NCOMMS-21-38671-T

Parental-fetal interplay of immune genes leads to fetal growth restriction

By:

Gurman Kaur et al.

Originality:

This study uses a unique humanized mouse model to study the role of HLA and KIR genes in fetal growth restriction.

Quality:

The overall quality of the paper is very good, and the methodology is both elegant and cutting edge. I would like to congratulate the team for their outstanding work. Despite an almost flawless setup, I have some major concerns that ought to be considered for acceptance in a high-impact journal.

1. As the authors clearly describe in the introduction, FGR can occur with and without preeclampsia. The role of HLA and KIR genes have been excessively studied in different ethnicity (reviewed by Moffett et al). Different combinations of these genes show a wide variation in risk depending on populations. What were the rationale to choose KIR2DI1 and HLA-C 0501? Please expand. Are these the optimal genes to study isolated FGR?

In the introduction, the authors describe different odds ratios for certain combinations. This part is lengthy and too detailed. Please describe the interactions of HLA and KIR genes in FGR and preeclampsia in a more pedagogic way so that the rationale for choosing the selected genes becomes understandable.

2. Placental sexual dimorphism is becoming a focus area in preeclampsia and FGR research. The placenta develops and functions differently in pregnancies with male or female fetuses. The authors need to include this aspect in the study and stratify the results based on fetal sex. This ought should be easily done considering that the fetuses have been genotyped. The Y-chromosome can be visualized by in-situ hybridization. This is an important concern that needs to be addressed to make the story complete.

3. The authors stimulated splenocytes from the transgenic mice. Were these splenocytes from pregnant mice or non-pregnant? It would be interesting to see how pregnancy per se alters their response in addition to the elegant controls performed.

4. Fetal weight was measured at gd18.5, was placenta weight and seize recorded at the same time?

5. The authors should be credited for taking on the effort to carefully study the blood pressure throughout pregnancy and to measure classical preeclampsia biomarkers. However, are the authors aware of any mice model where altered HLA/KIR genes give rise to preeclampsia features? Please discuss the specificity of the chosen genes in this perspective. Furthermore, the authors specifically altered the uterine NK cells. Could systemic effects such as preeclampsia, have been developed if all NK cells were altered, please discuss. The authors could consider breeding the females with STOX1 males to induce preeclampsia. The STOX 1 preeclampsia mouse model has been developed by D. Vaiman et al.

6. The evaluation of the spiral arteries was nicely performed. However, I do not understand why NK cells were collected at E9,5 and the spiral arteries evaluated a day later? Why not synchronize the days to better correlate the findings related to how NK cells prevent trophoblast invasion and spiral artery remodeling? Please explain.

7. Why did the authors not characterize the extravillous trophoblast placental trophoblasts by scRNA-seq the same way the other cell types were studied? Subtypes of EVT and their interaction with the NK cells would be interesting to decipher, particularly relating the findings on ECM and tissue remodeling.

8. The gene results are interesting and relevant for both preeclampsia and FGR. The authors need to complement their study with a histological description showing decidual and placental sections during the time for implantation and at the time of cell characterization E9,5 and E10,5. This could help visualize the maternal-fetal interface and help support the speculation on lymphocyte recruitment, oxidative stress, and apoptosis as well as their temporal aspects throughout early gestation.

9. In the discussion the authors state "These findings are of critical importance as they provide a rationale for why prevention or early intervention strategies are an important step in reducing long-term disease burden". What prevention or intervention are they referring to? Too unprecise and speculative, please be more specific.

Minor comments

1. Some minor spelling errors, please double-check.
2. Several sentences begin with an abbreviation, please change.

Reviewer #3 (Remarks to the Author):

In this manuscript, to demonstrate the hypothetical roles of KIR2DL1 and HLA-C*05 in fetal growth restriction (FGR) in human, authors made humanized mice that express KIR2DL1 on NK cells and its ligand HLA-C*05 on all MHC-class I expressing cells. All data indicate that expression of these genes were controlled as designed. The results supported the hypothetical roles of KIR2DL1 and HLA-C*05 in FGR. Advanced examination on them revealed that several classes of genes, which may be responsible for FGR. Findings in this study may provide a clue to ameliorate FGR in human and contribute to the healthy growth of children.

Introduction:

1. Introduction contains a large body of results and looks like the second abstract.

Results:

1. Fig. 3. It is understandable that authors mostly performed investigation on placenta and immune cells. However, because FGR increase perinatal mortality and morbidity, data on fetus/neonates, such as organ sizes and humoral factors of them and litter sizes, would attract broad attention including clinicians. These are a bit out of scope of this study, but such information could be valuable to understand impaired fetal/neonatal development of human.
2. Minor issue: Fig. 2b and Ext Data Fig. 1l; Please correct descriptions of the vertical axis because it can be read "IFN γ plus NK cells"

Reviewer #4 (Remarks to the Author):

The authors demonstrate that the interaction between two previously defined genes, the maternal KIR2DL1 expressed on uterine natural killer (NK) cells, and the paternally-inherited HLA-C*0501, expressed on fetal trophoblast cells, leads to FGR in a humanised mouse model. They further show that the initial KIR2DL1 and C*0501 interaction leads to pathogenic uterine arterial remodeling and a modulation of uterine NK cell function. They describe how this initial effect cascades to changes in transcriptional expression and intercellular communication in a myriad of cell types and pathways at the maternal-fetal interface using single cell sequencing and analysis. The authors ask an important question and leverage mouse models to dive into the biology behind a genetics finding. Overall the manuscript is well written and the figures are clear. This review focuses on the computational aspects of this work. The single cell sequencing and analysis are carried out in a robust way. The methodology that is used is appropriate and the visualizations are effective. The topic modeling approach with LDA is interesting, however traditional pathway and network analysis alone could be informative. It might also be interesting to consider trajectory analysis on the data to show transcriptional relationships between cell types.

Response to Reviewers

Overview

We thank the Reviewers for their appreciation of our work and their thoughtful suggestions and comments. We have addressed all the points in the revised manuscript with new data, analyses, and revisions to the text.

We summarize the key revisions below and then address each of the Reviewers' comments in the following point by point response.

The key highlights of our revision are:

- **Discussion of reasons underpinning the choice of risk genes in our model.**
- **Typing for the fetal Y chromosome using a Y-probe on purified fetal DNA to stratify fetal weight results based on sex.** We demonstrate that fetal sex has no impact on the manifestation of FGR in fetuses.
- **Histological examination of the placental/decidual tissue sections at mid-gestation (gd9.5) and additional histological data from placental sections at gd18.5.** We do not see evidence of any gross morphological changes between the different tissue sections obtained from the control and FGR matings.
- **Analysis of new data on litter sizes between the different mating groups.** We do not observe a significant difference in litter sizes between different mating groups.
- **Clarification of the analysis approaches used in our NK RNA-seq dataset.** We clarify and better describe the two approaches used (i.e. the traditional pathway and network analysis and topic modelling/LDA analysis) and highlight that the two approaches are complimentary.

- **Exploratory RNA velocity analysis on the unsorted cells dataset from the maternal-fetal interface.** We performed trajectory analysis on each sample in our unsorted cells RNA-seq dataset to explore the transcriptional relationships between cell types. We show that the results are consistent across genotypes but vary between the RNA velocity method used.

Response to Reviewer 1

In this manuscript by Kaur et al., the authors have set up an elegant system to study the contribution of interacting human immune genes HLA-C and KIR to fetal growth restriction (FGR). Genetic association studies in humans have implicated combinations of maternal KIR A haplotype and paternal group 2 HLA-C alleles with increased risk of pregnancy complications such as FGR. The interpretation is that inhibition of maternal natural killer (NK) cells by KIR2DL1, which binds to group 2 HLA-C alleles (C2) on fetal cells, restrains their pro-angiogenic function at the implantation site. However, the extensive polymorphism and complexity of both KIR and HLA genes make it difficult to tease apart the role of specific interacting gene pairs in FGR in humans.

*The authors exploited a humanised transgenic mouse model system to examine a single pair of HLA-C and KIR risk alleles. These are the KIR A haplotype KIR2DL1 and the group 2 HLA-C*0501. Remarkably, due to this interaction between an inhibitory KIR and its cognate HLA-C, they uncovered evidence of decreased vascular remodeling at the maternal fetal interface in mice during gestation. In addition to a ~12% decrease in weight, there were changes in spiral artery diameter and length that resulted in reduced vascular conductance and increased resistance to blood flow. It is noteworthy that FGR was observed only when there is expression of maternal KIR2DL1 and of the HLA-C*05 inherited paternally by the fetus.*

*They further carried out transcriptional profiling of this site in the mouse uterus to show that engagement of this receptor-ligand pair has consequences that impact gene programs relating to placentation not only in uterine NK (uNK) cells but also in other cell types at this site. Their single cell atlas at midgestation (gd9.5) identified 7 uNK subsets that showed features of increased lymphocyte activation and reduced ECM tissue remodeling programs. These FGR associated gene changes were not limited to uNK but were also seen in other cell types such as T and B lymphocytes, macrophages, stromal cells and trophoblasts. Despite significant differences in murine and human reproduction, such as the anatomy of implantation, this study highlighted surprising similarities between the human and murine maternal-fetal interface in terms of cellular crosstalk. They used the CellPhoneDB App to identify receptor-ligand pairs and showed a net increase in intercellular communication when KIR2DL1-HLA-C*0501 interactions occur.*

This is a carefully controlled study and the data are presented clearly. These findings have important implications for the study of HLA-KIR disease associations in general as they provide an in vivo correlate of human risk gene combinations in pregnancy that for obvious ethical reasons are impossible to understand at a mechanistic level in humans.

We thank the Reviewer for their appreciation of the importance and relevance of our findings, and for their thorough review of our manuscript.

Response to Reviewer 2

Originality: This study uses a unique humanized mouse model to study the role of HLA and KIR genes in fetal growth restriction.

We thank the Reviewer for acknowledging the novelty of the model we established and used for assessing fetal growth restriction in this study.

Quality: The overall quality of the paper is very good, and the methodology is both elegant and cutting edge. I would like to congratulate the team for their outstanding work. Despite an almost flawless setup, I have some major concerns that ought to be considered for acceptance in a high-impact journal.

We thank the Reviewer for their appreciation of the quality and approach used in our study and for their helpful and insightful comments. We have attempted to address the concerns raised by the Reviewer in the detailed response below.

1. As the authors clearly describe in the introduction, FGR can occur with and without preeclampsia. The role of HLA and KIR genes have been excessively studied in different ethnicity (reviewed by Moffett et al). Different combinations of these genes show a wide variation in risk depending on populations. What were the rationale to choose KIR2D11 and HLA-C 0501? Please expand. Are these the optimal genes to study isolated FGR?

In the introduction, the authors describe different odds ratios for certain combinations. This part is lengthy and too detailed. Please describe the interactions of HLA and KIR genes in FGR and preeclampsia in a more pedagogic way so that the rationale for choosing the selected genes becomes understandable.

We thank the Reviewer for this comment and regret any lack of clarity in our original description. Indeed, the role of HLA and KIR genes has been studied in different ethnicities

and there is more genetic diversity of the KIR locus (both at the allelic and haplotype level) in populations such as those from sub-Saharan Africa. However, despite this complexity, there is **consistency in the genetic risk association between the different populations, and the maternal KIR AA haplotype combined with a paternally-derived fetal HLA-C2 allele is still associated with the increased risk of pregnancy complications**, including FGR, pre-eclampsia and recurrent miscarriage (conditions which are thought to share the underlying problems associated with defective placentation)¹⁻⁴. The current genetic association datasets are not large enough to distinguish between FGR with or without pre-eclampsia.

Our choice of genes was driven by the composition and frequencies of alleles within the genetic risk haplotype. *KIR2DL1* was a good candidate because it is always present on the risk KIR A haplotype, and encodes for an inhibitory KIR with strict specificity for binding HLA-C2 ligands (part of the genetic risk gene combination). Amongst the different *KIR2DL1* alleles, *KIR2DL1*003* is the most common allele on the KIR A haplotype⁵. Correspondingly, we choose a common allele from the HLA-C2 group (*HLA-C*05*) which has been shown to bind *KIR2DL1*⁶, was expressed at relatively high levels in the human population, and therefore, was an example of a good representative allele from the HLA-C2 group.

We clarified this choice in the revised **Results (p. 5)**. We have also condensed the **Introduction** section about odds ratios to improve clarity (**p. 4**).

2. Placental sexual dimorphism is becoming a focus area in preeclampsia and FGR research. The placenta develops and functions differently in pregnancies with male or female fetuses. The authors need to include this aspect in the study and stratify the results based on fetal sex. This ought should be easily done considering that the fetuses have been genotyped. The Y-

chromosome can be visualized by in-situ hybridization. This is an important concern that needs to be addressed to make the story complete.

We thank the Reviewer for raising this important suggestion. To ask if there are any differences between male and female fetuses in the context of FGR, we have typed for the fetal Y chromosome using a Y-probe on purified fetal DNA collected at gd18.5, while assessing FGR. We show that **fetal sex has no impact on the manifestation of FGR (fetal weight) in fetuses from the KIR x C*05 FGR mating group.** (Fig. R1). Furthermore, the proportion of KIR x C*05 fetuses that were below the 5th or the 10th birth weight centile of the control were statistically indistinguishable between male and female fetuses (Fig. R2). We note this in the revised manuscript (p. 9).

Fig. R1. Fetal weight determined at gd18.5 stratified by fetal sex of progeny from the KIR x C*05 FGR mating combination. Mean \pm SEM is shown. n = 86 (data from 11 different litters).

Fig. R2. Fetal weight proportions from the KIR x C*05 FGR mating compared to the WT x C*05 control mating. Percent of male and female fetuses are shown at different ranges of weight quantiles of the WT x C*05 mating controls.

3. *The authors stimulated splenocytes from the transgenic mice. Were these splenocytes from pregnant mice or non-pregnant? It would be interesting to see how pregnancy per se alters their response in addition to the elegant controls performed.*

We regret the lack of clarity on this point. We performed the splenocyte stimulation experiments on **non-pregnant mice**. These were part of characterising our humanised transgenic mouse model to validate whether the human transgenes were functional in the mouse model. We agree with the Reviewer that it is indeed interesting to understand how pregnancy alters stimulation responses compared to non-pregnant females. This has been a subject of investigation in several other research articles referenced below⁷⁻¹⁰.

4. *Fetal weight was measured at gd18.5, was placenta weight and seize recorded at the same time?*

Yes, this is correct. Both fetal and placental weight measurements were performed at gd18.5. We clarified this in the revised **Results** (p. 9).

5. The authors should be credited for taking on the effort to carefully study the blood pressure throughout pregnancy and to measure classical preeclampsia biomarkers. However, are the authors aware of any mice model where altered HLA/KIR genes give rise to preeclampsia features? Please discuss the specificity of the chosen genes in this perspective. Furthermore, the authors specifically altered the uterine NK cells. Could systemic effects such as preeclampsia, have been developed if all NK cells were altered, please discuss. The authors could consider breeding the females with STOX1 males to induce preeclampsia. The STOX 1 preeclampsia mouse model has been developed by D. Vaiman et al.

We thank the Reviewer for acknowledging our effort in measuring the blood pressure and classical biomarkers of pre-eclampsia, and for these interesting questions. We are not aware of any other mouse model that has studied HLA/KIR gene responses in the context of pre-eclampsia. Nevertheless, as discussed in the response to comment 1 above, genetic association studies in humans do point towards an increased risk of developing pre-eclampsia in pregnancies involving a KIR AA mother and an HLA-C2 father. This motivated our choice of KIR2DL1 and HLA-C*05 genes (now noted in **p. 4** of the revised **Introduction** and **p.5** of the revised **Results**).

Moreover, in our model, **all NK cells in the mouse expressed KIR2DL1 (Fig. 1h,i)** such that the genetic alteration was not specific to uterine NK cells alone. However, we did not observe any features of pre-eclampsia. One possible explanation is that manifestation of pre-eclampsia

requires additional risk factors or mechanisms on top of those involved in FGR. We note this in the revised **Results (p. 10)**.

As the Reviewer notes, in the STOX1 pre-eclampsia model, a mouse model which overexpresses the transcription factor human STOX1, WT females mated with STOX1 overexpressing transgenic males manifest symptoms of pre-eclampsia. Interestingly, the increase in systolic blood pressure in these matings was seen almost immediately after the mating occurred, suggesting that onset of these symptoms was independent of placental development¹¹. There is no evidence of interaction between STOX1 and KIR and HLA genes. As a result, mating KIR2DL1-expressing females with STOX1 overexpressing males is unlikely to reveal new mechanistic insight into the role of KIR2DL1-HLA-C*05 genes in inducing FGR or pre-eclampsia, and is beyond the scope of this study.

6. The evaluation of the spiral arteries was nicely performed. However, I do not understand why NK cells were collected at E9,5 and the spiral arteries evaluated a day later? Why not synchronize the days to better correlate the findings related to how NK cells prevent trophoblast invasion and spiral artery remodelling? Please explain.

We thank the Reviewer for this question.

First, we collected uNK cells at mid-gestation as this correlated with the kinetics of uNK cell accumulation at the maternal-fetal interface. Mid-gestation (gd9.5) represented the time when the frequency of uNK cells peaks at the maternal-fetal interface, as documented by previous studies^{12,13}. We clarify this in the revised manuscript (**p. 8, 11**).

Next, following our analysis which revealed disease-relevant RNA expression programs at gd9.5, we wanted to investigate the effects of these expression programs on the structural aspects of spiral artery development and vascular remodelling. **gd10.5 represented the time when the large-diameter coiling spiral arteries have formed in the decidua**, and supply blood flow to the maternal canals, as demonstrated by previous scanning electron microscope studies^{14,15}. Because we wanted to investigate the effects on the structural features of spiral artery development and also measure the resistance to maternal blood flow through the entire uteroplacental vascular network, it was important to do it at a biologically-relevant timepoint when the uteroplacental circulation has fully developed. Therefore, the assessment of spiral arteries and the uteroplacental vasculature was performed at gd10.5. We clarify this in the revised manuscript (**p. 10**).

7. Why did the authors not characterize the extravillous trophoblast placental trophoblasts by scRNA-seq the same way the other cell types were studied? Subtypes of EVT's and their interaction with the NK cells would be interesting to decipher, particularly relating the findings on ECM and tissue remodeling.

We regret the lack of clarity on this point. **We have characterised the different populations of trophoblast cells by scRNA-seq along with all other cells from the maternal-fetal interface at mid-gestation (Fig. 7b)**. As there are no reliable cell surface markers of trophoblasts, we collected total unsorted live cells from the maternal-fetal interface, profiled them by scRNA-seq and used unsupervised clustering and *post-hoc* annotation to analyse the data. We observed FGR-associated expression changes in different subsets of trophoblasts in genes involved in invasion, angiogenesis and ECM remodelling amongst others (**Fig. 7c and Extended Data Fig. 8j**). We also present changes in interaction between NK cells and

trophoblast subsets, and highlight specific receptor-ligand interaction pairs that are altered in disease (**Fig. 7d-f**).

8. The gene results are interesting and relevant for both preeclampsia and FGR. The authors need to complement their study with a histological description showing decidual and placental sections during the time for implantation and at the time of cell characterization E9,5 and E10,5. This could help visualize the maternal-fetal interface and help support the speculation on lymphocyte recruitment, oxidative stress, and apoptosis as well as their temporal aspects throughout early gestation.

We thank the reviewer for appreciating the relevance of our findings and for this suggestion. We analysed sections from fresh-frozen tissue samples from implantation sites at **gd9.5 from both control and FGR matings**. To validate findings from specific genes and gene programs altered in FGR, we performed *in-situ* hybridisation for genes highlighted via topic modelling and differential gene expression, along with a counter-stain of placental/decidual tissue sections with hematoxylin, which allowed for histological examination of these tissue sections.

We analysed the obtained images from these placental/decidual and demonstrated changes in gene expression of *Ccl1*, *Gzme/d/g* and *Anxa2* in the implantation site sections obtained from FGR matings (**Fig. 6d,e and Extended Data Fig. 8h**). Notably, we did not see any evidence of gross morphological changes between the different tissue sections obtained from WT x Cw5 control and KIR x Cw5 FGR matings. To investigate additional effects on placental development, we also examined placental tissue sections obtained from WT and FGR matings at **gd18.5 (at the time of assessment of FGR)**, and again did not observe any

evidence of gross histological changes in placental tissue sections between WT and FGR matings (in line with similar placental weight across the different mating combinations shown in **Fig. 3c**) (**Fig. R3**).

Fig. R3. Histological stainings from placenta (gd18.5) from control (WT x C*05, left panels) and FGR (KIR x C*05, right panels) matings.

9. In the discussion the authors state “These findings are of critical importance as they provide a rationale for why prevention or early intervention strategies are an important step in reducing long-term disease burden”. What prevention or intervention are they referring to? Too unprecise and speculative, please be more specific.

We thank the Reviewer for pointing this out. Our work reveals new mechanistic insight into the role of specific KIR and HLA genes in mediating FGR at a critical timepoint early in gestation. We also highlight pathways, genes and specific receptor-ligand pairs that are altered in disease. Targeted interference in these disease-specific functions or receptor-ligand interactions have potential for therapeutic intervention. In addition, our work also paves the

way towards a diagnostic screening process to identify pregnancies at high risk of developing FGR, which could benefit from better monitoring and early clinical intervention, thereby offsetting the long-term risk of serious health implications. We have rephrased the sentence in the **Discussion** accordingly (p. 20).

Minor comments

1. *Some minor spelling errors, please double-check.*
2. *Several sentences begin with an abbreviation, please change.*

We thank the Reviewer for pointing these out and have addressed these comments and made sure that each abbreviation has been defined the first time it appears in the text.

Response to Reviewer 3

*In this manuscript, to demonstrate the hypothetical roles of KIR2DL1 and HLA-C*05 in fetal growth restriction (FGR) in human, authors made humanized mice that express KIR2DL1 on NK cells and its ligand HLA-C*05 on all MHC-class I expressing cells. All data indicate that expression of these genes were controlled as designed. The results supported the hypothetical roles of KIR2DL1 and HLA-C*05 in FGR. Advanced examination on them revealed that several classes of genes, which may be responsible for FGR. Findings in this study may provide a clue to ameliorate FGR in human and contribute to the healthy growth of children.*

We thank the Reviewer for their thorough review of our manuscript, helpful suggestions, and appreciation of our work.

Introduction:

1. Introduction contains a large body of results and looks like the second abstract.

We appreciated the Reviewer's comment and have now condensed the summary of the results in the **Introduction (p. 4)**.

Results:

1. Fig. 3. It is understandable that authors mostly performed investigation on placenta and immune cells. However, because FGR increase perinatal mortality and morbidity, data on fetus/neonates, such as organ sizes and humoral factors of them and litter sizes, would attract broad attention including clinicians. These are a bit out of scope of this study, but such information could be valuable to understand impaired fetal/neonatal development of human.

We agree with the Reviewer that data on fetal/neonatal development such as organ sizes or humoral factors would be of interest to the larger clinical community, although, as the Reviewer notes this does fall outside of the scope of this study. Note, we did collect and analyse data on litter sizes, and did not observe any significant difference between the different mating groups. We added this in the revised manuscript (**p. 10**). We agree that further data on neonates would be an important point for future studies and note this in the revised **Discussion (p. 23)**.

2. Minor issue: Fig. 2b and Ext Data Fig. 1l; Please correct descriptions of the vertical axis because it can be read "IFN γ plus NK cells"

We thank the reviewer for catching this. We have corrected these descriptions.

Response to Reviewer 4

*The authors demonstrate that the interaction between two previously defined genes, the maternal KIR2DL1 expressed on uterine natural killer (NK) cells, and the paternally-inherited HLA-C*0501, expressed on fetal trophoblast cells, leads to FGR in a humanised mouse model. They further show that the initial KIR2DL1 and C*0501 interaction leads to pathogenic uterine arterial remodeling and a modulation of uterine NK cell function. They describe how this initial effect cascades to changes in transcriptional expression and intercellular communication in a myriad of cell types and pathways at the maternal-fetal interface using single cell sequencing and analysis. The authors ask an important question and leverage mouse models to dive into the biology behind a genetics finding. Overall the manuscript is well written and the figures are clear. This review focuses on the computational aspects of this work. The single cell sequencing and analysis are carried out in a robust way. The methodology that is used is appropriate and the visualizations are effective.*

We thank the Reviewer for their appreciation of our work, both in terms of biological experiments and insights, and in terms of computational analysis and visualisation.

The topic modeling approach with LDA is interesting, however traditional pathway and network analysis alone could be informative.

We thank the Reviewer for raising this question about the utility of traditional pathway and network analysis in addition to LDA. We **present results from both analysis approaches in our manuscript**, as follows:

- (1) We performed analysis of differentially expressed genes and pathways between cell clusters/subsets, using the datasets from sorted NK cells (**Fig. 5d, Fig. 6f, Supplementary Table 5 and Supplementary Table 12**) as well as on total (unsorted) cells from the maternal-fetal interface (**Fig. 7b,c, Extended Data Fig. 8i,j, Supplementary Table 7, and Supplementary Table 13**).
- (2) We performed topic modelling with latent Dirichlet allocations on the sorted NK cell data (**Fig. 6b,c, Extended Data Fig. 7 and Supplementary Table 4**).

We find that these approaches are often complementary. The differentially expressed genes/pathway analysis approach allows us to focus on differences between cell subsets and interpret them in the light of well-established knowledge. However, this may not suffice in this system, because, especially NK cell profiles follow a continuum of cell states (not readily partitioned to discrete subsets) and moreover, cells from different subsets could activate the same gene program. We have encountered such circumstances before in both innate lymphoid cells (ILCs)¹⁶ and other immune cells¹⁷, and in such instances, topic modelling by LDA is better suited to recover the relevant programs in the data (which can still be tested *post hoc* for enriched pathways).

Notably, in both our sorted NK and unsorted datasets, we also performed gene set enrichment analysis using *enrichR*^{18,19} and pathway analysis using Ingenuity pathway analysis software (**Extended Data Fig. 8i,j and Supplementary Table 8**). In some cases, FGR-relevant genes were not fully curated in available pathway analysis tools, and we augmented these with manually curated gene sets and functional annotations of disease-relevant genes.

We have revised our manuscript to clarify the two analysis approaches used (p. 12, 14, 15, 16, 18).

It might also be interesting to consider trajectory analysis on the data to show transcriptional relationships between cell types.

We thank the Reviewer for this interesting suggestion, which we have now explored by RNA velocity analysis using both the steady state and dynamic models in scVelo²⁰ (**Fig. R4**), conducted separately on a per sample basis for each genotype mating combination. The results were generally consistent between the genotype mating combinations.

In the **stromal subsets**, the analysis suggests two differentiation paths: (1) from the least differentiated stromal subsets 2, 3, and 4 (S2, S3, S4) through subsets S6 and S1 to endpoint subset S5, and (2) S2 and/or S4 to the stromal fibroblast subset. The velocity arrows in S2, S3, and S4 change slightly depending on whether the steady state or dynamic model was used for the analysis.

In the **NK subsets**, the two models (steady state and dynamic) are incongruent with reversal of velocity direction. The steady state model supports a path from trNKs to the mixed cNK/trNK subset; in the dynamic model, there is a path from the mixed cNK/trNK subset to the trNK subset.

A recent preprint²¹ analysed the performance and assumptions of RNA velocity approaches, highlighting discrepancies and discussing the suitability of underlying assumptions. In light of this, and given the discrepancies we have observed between the steady state and dynamic

models, we decided not to include these results in the current manuscript. Absent a ground truth, we cannot justify the choice of one model over another. We have mentioned the different stages of stromal cell differentiation/decidualisation in the manuscript, which was also evident using traditional marker gene analysis (p. 17).

Fig. R4. Trajectory analysis of total unsorted cells from the maternal-fetal interface using steady state (left panel) and dynamic (right panel) models available in scVelo. Displayed is a representative figure from cells isolated

from a control mating combination i.e. WT x C*05 (43_1H) and a FGR mating combination i.e. KIR x C*05 (17_3D). The cell subset annotations are as described in Fig. 7b.

References:

- 1 Hiby, S. E. *et al.* Maternal KIR in combination with paternal HLA-C2 regulate human birth weight. *J Immunol* **192**, 5069-5073, doi:10.4049/jimmunol.1400577 (2014).
- 2 Hiby, S. E. *et al.* Maternal activating KIRs protect against human reproductive failure mediated by fetal HLA-C2. *J Clin Invest* **120**, 4102-4110, doi:10.1172/JCI43998 (2010).
- 3 Hiby, S. E. *et al.* Combinations of maternal KIR and fetal HLA-C genes influence the risk of preeclampsia and reproductive success. *J Exp Med* **200**, 957-965, doi:10.1084/jem.20041214 (2004).
- 4 Nakimuli, A. *et al.* A KIR B centromeric region present in Africans but not Europeans protects pregnant women from pre-eclampsia. *Proc Natl Acad Sci U S A* **112**, 845-850, doi:10.1073/pnas.1413453112 (2015).
- 5 Bashirova, A. A., Martin, M. P., McVicar, D. W. & Carrington, M. The killer immunoglobulin-like receptor gene cluster: tuning the genome for defense. *Annu Rev Genomics Hum Genet* **7**, 277-300, doi:10.1146/annurev.genom.7.080505.115726 (2006).
- 6 Colonna, M., Borsellino, G., Falco, M., Ferrara, G. B. & Strominger, J. L. HLA-C is the inhibitory ligand that determines dominant resistance to lysis by NK1- and NK2-specific natural killer cells. *Proc Natl Acad Sci U S A* **90**, 12000-12004, doi:10.1073/pnas.90.24.12000 (1993).

- 7 Veenstra van Nieuwenhoven, A. L. *et al.* Endotoxin-induced cytokine production of monocytes of third-trimester pregnant women compared with women in the follicular phase of the menstrual cycle. *Am J Obstet Gynecol* **188**, 1073-1077, doi:10.1067/mob.2003.263 (2003).
- 8 Faas, M. M. *et al.* Porphyromonas Gingivalis and E-coli induce different cytokine production patterns in pregnant women. *PLoS One* **9**, e86355, doi:10.1371/journal.pone.0086355 (2014).
- 9 Sacks, G. P., Redman, C. W. & Sargent, I. L. Monocytes are primed to produce the Th1 type cytokine IL-12 in normal human pregnancy: an intracellular flow cytometric analysis of peripheral blood mononuclear cells. *Clin Exp Immunol* **131**, 490-497, doi:10.1046/j.1365-2249.2003.02082.x (2003).
- 10 Luppi, P. *et al.* Monocytes are progressively activated in the circulation of pregnant women. *J Leukoc Biol* **72**, 874-884 (2002).
- 11 Doridot, L. *et al.* Preeclampsia-like symptoms induced in mice by fetoplacental expression of STOX1 are reversed by aspirin treatment. *Hypertension* **61**, 662-668, doi:10.1161/HYPERTENSIONAHA.111.202994 (2013).
- 12 Lima, P. D., Croy, B. A., Degaki, K. Y., Tayade, C. & Yamada, A. T. Heterogeneity in composition of mouse uterine natural killer cell granules. *J Leukoc Biol* **92**, 195-204, doi:10.1189/jlb.0312136 (2012).
- 13 Sojka, D. K. *et al.* Cutting Edge: Local Proliferation of Uterine Tissue-Resident NK Cells during Decidualization in Mice. *J Immunol* **201**, 2551-2556, doi:10.4049/jimmunol.1800651 (2018).
- 14 Adamson, S. L. *et al.* Interactions between trophoblast cells and the maternal and fetal circulation in the mouse placenta. *Dev Biol* **250**, 358-373, doi:10.1016/s0012-1606(02)90773-6 (2002).

- 15 Rennie, M. Y. *et al.* in *The Guide to Investigation of Mouse Pregnancy* (eds B. Anne Croy, Aureo T. Yamada, Francesco J. DeMayo, & S. Lee Adamson) 201-210 (2014).
- 16 Bielecki, P. *et al.* Skin-resident innate lymphoid cells converge on a pathogenic effector state. *Nature* **592**, 128-132, doi:10.1038/s41586-021-03188-w (2021).
- 17 Xu, H. *et al.* Transcriptional Atlas of Intestinal Immune Cells Reveals that Neuropeptide alpha-CGRP Modulates Group 2 Innate Lymphoid Cell Responses. *Immunity* **51**, 696-708 e699, doi:10.1016/j.immuni.2019.09.004 (2019).
- 18 Chen, E. Y. *et al.* Enrichr: interactive and collaborative HTML5 gene list enrichment analysis tool. *BMC Bioinformatics* **14**, 128, doi:10.1186/1471-2105-14-128 (2013).
- 19 Kuleshov, M. V. *et al.* Enrichr: a comprehensive gene set enrichment analysis web server 2016 update. *Nucleic Acids Res* **44**, W90-97, doi:10.1093/nar/gkw377 (2016).
- 20 Bergen, V., Lange, M., Peidli, S., Wolf, F. A. & Theis, F. J. Generalizing RNA velocity to transient cell states through dynamical modeling. *Nat Biotechnol* **38**, 1408-1414, doi:10.1038/s41587-020-0591-3 (2020).
- 21 Gorin, G., Fang, M., Chari, T. & Pachter, L. RNA velocity unraveled. *bioRxiv*, 2022.2002.2012.480214, doi:10.1101/2022.02.12.480214 (2022).

REVIEWERS' COMMENTS

Reviewer #1 (Remarks to the Author):

We had already endorsed the manuscript in the previous review. After revision and replies to comments from other reviewers, the manuscript is even stronger.

Reviewer #2 (Remarks to the Author):

The authors have done a very good job in revising the manuscript and I would like to congratulate them on an excellent paper. I have no further comments.

Reviewer #3 (Remarks to the Author):

Authors have addressed all my concerns, which include additional data that may not be necessary but attract readers interest.
So this reviewer has no further comments.

Reviewer #4 (Remarks to the Author):

The authors have addressed my concerns.

Reviewers' comments

Reviewer 1

We had already endorsed the manuscript in the previous review. After revision and replies to comments from other reviewers, the manuscript is even stronger.

Reviewer 2

The authors have done a very good job in revising the manuscript and I would like to congratulate them on an excellent paper. I have no further comments.

Reviewer 3

Authors have addressed all my concerns, which include additional data that may not be necessary but attract readers interest.

So this reviewer has no further comments.

Reviewer 4

The authors have addressed my concerns.

Response to Reviewers' comments

We thank the Reviewers for their appreciation of our work, their helpful and supportive comments, and for a thorough review of our manuscript.